



# Ice nucleating ability of particulate emissions from solid biomass-fired cookstoves: an experimental study

Kimmo Korhonen[1], Thomas Bjerring Kristensen[2], John Falk[2], Robert Lindgren[3], Christina Andersen[4],

Ricardo Luis Carvalho[3,a], Vilhelm Berg-Malmborg[4], Axel Eriksson[4], Christoffer Boman[3], Joakim

Pagels[4], Birgitta Svenningsson[2], Mika Komppula[5], Kari E.J. Lehtinen[1] and Annele Virtanen[1]

[1]University of Eastern Finland, Dept. Applied Physics. P.O. box 1627, FI-70211 Kuopio, Finland

[2]Lund University, Department of Physics, SE-22100, Lund, Sweden

[3]Umeå University, Thermochemical Energy Conversion Laboratory, SE-90187, Umeå, Sweden

[4]Lund University, Ergonomics and Aerosol Technology, Box 118, Lund SE-22100, Sweden

[5]Finnish Meteorological Institute, Atmospheric Research Centre of Eastern Finland, P.O. box 1627, FI-70211 Kuopio, Finland

[a] Now at: Centre for Environmental and Marine Studies, University of Aveiro, Department of Environment and Planning, PT-3810-193, Aveiro, Portugal

*Correspondence to*: Kimmo Korhonen (Kimmo.Korhonen@uef.fi)

**Abstract.** Ice nucleating abilities of particulate emissions from solid-fuel burning cookstoves were studied using a portable ice nuclei counter SPIN (**SP**ectrometer for **I**ce **N**uclei) as part of the SUSTAINE (**S**alutary **U**meå **ST**udy of **A**erosols **IN** Biomass Cookstove **E**missions) laboratory experiment campaign. The emissions were generated from two traditional cookstove types commonly used for household cooking in sub-Saharan Africa, and two advanced gasifier stoves which are under research to promote sustainable development alternatives. The studied solid fuels included biomass from two different African tree species, Swedish softwood and agricultural residue products relevant to the region. Measurements were performed with a modified version of the standard water boiling test on 1) polydisperse samples from flue gas during burning and 2) size-selected accumulation mode (250-500 nm) soot particles from a 15-m$^3$ aerosol-storage chamber, from which the particles were introduced to water-supersaturated freezing conditions in the SPIN.

We observed that accumulation mode soot particles generally produced an ice-activated fraction of $10^{-3}$ in temperatures that were 1-1.5 $^{\circ}$C higher than what was required for homogeneous freezing at fixed $RH_{water}$ = 115 %. Five special experiments where the combustion performance of one cookstove was intentionally modified were also performed, which led to a significant increase in the ice nucleating ability of the particles in two experiments, resulting in $10^{-3}$ ice activation at up to 5.9

$^O$C higher temperatures than homogeneous freezing. Moreover, six different physico-chemical properties of the emission particles were investigated but we did not find a clear correlation between them and increasing ice-nucleating ability. We conclude that in general, the studied freshly emitted combustion aerosols only facilitate immersion freezing at temperatures moderately above where homogeneous freezing occurs.

## 1 Introduction

Mixed-phase clouds (MPCs) play an essential role for climate and the hydrological cycle. Cloud droplets freeze homogeneously at temperatures near -38$^O$C, but ice nucleating particles (INPs) may catalyze freezing of supercooled cloud droplets at higher temperatures. MPC properties and lifetime are very sensitive to the formation of solid ice, but there are significant gaps in our knowledge regarding these important ice formation processes.

INPs active in immersion freezing at a temperature of -30$^O$C are relatively rare in the lower troposphere with concentrations at the order of 0.01 cm$^{-3}$ in many regions (DeMott et al., 2010). Ambient INPs include dust and biological particles and potentially soot particles (Hoose and Möhler, 2012, and references therein). Soot particles from an acetylene burner, a kerosene burner and a soot generator have been reported to be active in immersion freezing at temperatures up to -24$^O$C (DeMott, 1990), -20$^O$C (Diehl and Mitra, 1998) and -10$^O$C (Gorbunov et al., 2001), respectively. However, a wide range of soot particles have been reported to be inefficient as INPs in immersion mode (e.g. Hoose and Möhler, 2012), and the available parameterisations of the ice nucleating ability of soot particles span several orders of magnitude (Vergara-Temprado et al., 2018). The radiative forcing associated with the impact of soot particles on MPCs has been reported to be very uncertain and potentially up to about 1 Wm$^{-2}$ (Yun et al., 2013). The Intergovernmental Panel on Climate Change (IPCC) expressed in their latest assessment report a great need for additional research with respect to the role of soot particles in heterogeneous ice nucleation (Boucher et al., 2013).

Soot particles are produced from incomplete combustion and ambient soot particle properties are highly variable and influenced by the combustion conditions and atmospheric ageing (Corbin et al., 2015). It is not entirely clear which soot particle properties influence the ice nucleating ability of the particles. Chemical groups on the soot particle surface with the ability to form hydrogen bonds with water molecules are likely to be of importance (Gorbunov et al., 2001) as well as the soot particle nanostructure (Häusler et al., 2018).

Biomass combustion for cooking is an important global source of energy, being the major environmental health risk worldwide, as it is a significant source of particulate matter (PM) and soot particles on regional to global scales (Lim et al., 2012; Bonjour et al., 2013). However, studies of the associated ice nucleating ability are scarce. Ambient measurements indicate that biomass combustion is a source of INPs for MPC conditions (Twohy et al., 2010; McCluskey et al., 2014).


Detectable condensation/immersion freezing INP concentrations for a temperature of -30 $^O$C have been reported for simulated wildfires in 9 of 21 and 13 of 22 experiments, respectively (Petters et al., 2009; Levin et al., 2016). Refractory

black carbon has been associated with a significant fraction of the emitted INPs (Levin et al., 2016). Combustion with a modern log wood burner produced fresh emissions of INPs for a temperature of -35$^O$C but not for a temperature of -30$^O$C for condensation/immersion freezing (Chou et al., 2013). Huang et al. (2018) modeled the indirect climate impact of solid fueled cookstove aerosol emissions, and they reported a potentially significant climate impact from emitted soot particles on MPC conditions and thus climate. The most common way of using biomass is still in traditional cookstoves/open fires, so called 3-

stone fires, used in poorly ventilated spaces, resulting in severe emissions of air pollutants. However, an efficient utilization of modern biomass fuels in efficient biomass cookstoves can constitute an alternative to mitigate climate-forcing aerosol emissions and household air pollution (HAP) worldwide (Carvalho et al. 2016). In Sub-Saharan Africa, the extensive use of firewood and charcoal also leads to considerable forest and soil degradation. Thus, densification of various locally produced biomass feedstocks and residues in the form of fuel pellets can contribute to a more sustainable utilization of biomass in

households when comparing with the traditional harvesting and combustion of wood from natural forests (Carvalho et al. 2019). Additionally, the combustion of biomass pellets in micro-gasifier cookstoves appears as a relatively clean solution, and their emission performance can be similar to that achieved by gas stoves (Champion and Grieshop, 2019).

In this study, we present the first measurements of the ice nucleating ability of freshly emitted aerosol particles from

biomass-fired cookstoves. We investigated the aerosol emissions from four different cookstoves representing different advances in technology and using a wide range of biomass fuel types. The ice nucleating ability of aerosol particles generated during standard water boiling tests under well-controlled laboratory conditions was measured online with a continuous flow diffusion chamber (CFDC) during the **S**alutary **U**meå **ST**udy of **A**erosols **IN** Biomass Cookstove **E**missions (SUSTAINE) campaign.

**2 Experiment methodology**

**2.1 Experiment set-up**

Cooking simulations were carried out at the Thermochemical Energy Conversion Laboratory (TEC-lab) of Umeå University, and the sampling set-up and relevant supporting instrumentation for the ice nucleation (IN) experiments are presented in Fig. 1. The emissions were generated using a modified version of the standardized Water Boiling Test (WBT) version 4.2.4,

where the cookstoves and fuels were used to heat up 5 liters of water from room temperature in a metal cooking pot. A typical simulation time was 60-85 minutes in total, including two phases of the WBT process: cold start (15-40 min) and simmering (45 min). The cooking pot was set to a designated height for each cook stove used in all but three special experiments. Solid biomass fuels were lit using 12 g of ethanol in each experiment, and water vapor from the boiling was diverted away from the sample collector dome to prevent excess humidity entering the sampling lines. The sample aerosol





was diluted to 1:100, cooled down to room temperature and dried to < 10% $RH_w$ using ejector dilution (marked "ED" in Fig. 1) on dry and filtered compressed air to avoid saturation of analyzers. The supportive data were collected using multiple instruments in parallel with the CFDC instrument. Particle size distribution was monitored during the WBT using a Fast Particle Analyzer (FPA, Cambustion DMS 500) that was connected to the transient line "A" in Fig. 1. The FPA allowed real-time monitoring of the particle size distribution with its data output resolution of 10 Hz. Other instruments in the line "A"

were a high-resolution time-of flight soot particle aerosol mass spectrometer (SP-AMS, Aerodyne Inc.) for measuring the physico-chemical composition of the aerosol particles, and a seven-wavelength aethalometer (Magee Scientific AE33, sampling rate 1 Hz) measuring black carbon (BC) concentrations. The two latter instruments were connected interchangeably to the chamber sampling line "B" (Fig. 1) for chamber experiments. An SMPS system (classifier, TSI 3082 + condensation particle counter, TSI 3775) measured particle size distribution and concentration in chamber experiments at

scan range from 15.7 nm to 615.3 nm, with continuously repeated 180-second scans during aerosol injection and sampling. Other instruments used in chamber experiments were a cloud condensation nuclei counter (CCNC, Droplet Measurement Technologies) that sampled in parallel with an aerosol particle mass analyzer (APM, Kanomax) downstream of a differential mobility analyzer (TSI model 3071, marked DMA$_{CCN}$ in Fig. 1) whose voltage was systematically varied for obtaining APM and CCNC scans at mobility diameters of 65 nm, 100 nm, 200 nm and at times also 350 nm. The time resolution of this

measurement was approximately seven minutes between changes in DMA$_{CCN}$ voltage, which enabled obtainment of one APM spectrum and 1-2 full scan cycles on the CCNC.

The set-up allowed two different types of IN experimentation: transient and chamber experiments. The transient ones were conducted during the cooking simulation where for this purpose sample line "A" of Fig. 1 was used. The purpose of transient

experiments was to study the possible IN potential from fresh, and polydisperse cookstove emissions during the simmering phase of the WBT. The duration of each cooking simulation was at least 30 minutes (simmering phase), which enabled one scan on the CFDC, over an $RH$ range for one sampling temperature. A more detailed description of the CFDC experiments is presented in chapter 2.3. The chamber experiments were carried out to extend sampling time up to slightly more than two hours, and for this polydisperse aerosol was ejected into the chamber after preliminary drying and dilution so the sampling

could be continued after the end of the WBTs. The chamber used in the experiments was a 15-m$^3$ stainless steel aerosol storage chamber. The chamber was purged with filtered air prior to each experiment. It was filled following the sampling set-up of Fig. 1 to a mass concentration of 10-100 µg m$^{-3}$ before the IN experimentation with the CFDC was started. A typical aerosol injection time that was required to reach sufficient mass concentration in the chamber was 10-40 minutes, which represents one full combustion cycle. After filling, the typical sampling time was ca. 2 hours before the background

signal of the CFDC increased too high or deposition effects, such as coagulation and wall losses, reduced the aerosol concentration too much for optimal sampling. Deposition through wall losses was strongest for ultrafine particle sizes due to diffusion, and longer sampling times were possible for accumulation mode particles only.



A CPC was used in parallel to the CFDC and it recorded typical concentrations of 30-200 cm⁻³ at the 250-500 nm size range

after size-selection of the sample particles. The focus of the chamber experiments was to investigate the role of accumulation mode soot particles as INPs, and the samples were size-selected using a Vienna type DMA (DMA$_{IN}$ in Fig. 1) prior to the CFDC and the CPC, aiming for the largest particle size that was present with a reasonable number concentration (>30 cm⁻³). It is worth emphasizing that the chamber was used only for storing and mixing aerosol potentially representing different combustion phases and no simulated atmospheric aging via e.g. oxidation or interaction with other chemicals was applied to

any emission samples that were studied using the CFDC.

## 2.2 Biomass feedstocks and cookstoves

The fuels used in this study consist of firewood species available in large areas of sub-Saharan Africa and commonly used in residential cooking, and regionally relevant agricultural residues. The *Casuarina equisetifolia* (CAS), commonly known as Horsetail tree, is a species which is commonly used in agroforestry systems in the southern part of the African continent. In

this work, the CAS was selected because it is a tree species that can be used in the implementation of sustainable agroforestry systems in East Africa. According to the Food and Agriculture Organization (FAO), agroforestry is a powerful tool in enhancing various ecosystem services, including the enhancement of food productivity, landscape regeneration and woody biomass production (Food and Agriculture Organization of the United Nations, 2017). Furthermore, the *Sesbania sesban* (SES) is also a tree applied to support the implementation of agroforestry systems, being native in most parts of sub-

Saharan Africa and widespread in Kenya where the African fuels were collected for this study.

In addition to firewood, four types of agricultural residues were studied. Africa produces annually approximately one million tons of coffee (International Coffee Organization, 2018), the largest coffee exporting countries being located in sub-Saharan regions of the continent. During processing, agricultural residues form up to 50% of the total weight of coffee products

(Oliveira and Franca, 2015) and the usage of these residues as biofuels is under research for potential sustainable development in sub-Saharan Africa. The coffee husk (CH) used in this study were collected from Kenya where *Coffea arabica* is the dominant coffee species. Moreover, bagasse (BG) is a by-product of sugar production which has also demonstrated to be suitable for bioenergy production. This residue is usually disposed when the juice is extracted from sugarcanes. In this study, we focus on the utilization of sugarcane bagasse for biomass fuel production and use, as sugarcane

(*Saccharum officinarum*) is an important agricultural product in most parts of Sub-Saharan African countries and the biomass potential of this residue is also studied to promote sustainability. Additionally, the Water Hyacinth (WH), *Eichhornia crassipes*, is an indigenous species from South America, but it was introduced to Africa during the colonial period. It is a very invasive species, which requires constant population control, hence its utilization is proposed for energy purposes. Consequently, the removal of some of these plants from African lakes contributes to the sustainability of these

water systems, producing significant amounts of combustible waste whose potential as a biofuel is also under research.



Furthermore, rice husk (RC) is an agricultural residue from processing African rice, *Oryza glaberrima,* to cereal products. Rice production has been growing steadily in Africa for decades and processing the plants from paddy rice to milled rice produces approximately 25 weight percent of husks (Muthayya *et al.*, 2014), whose potential as a biofuel like BG is being
studied for sustainable energy use. Finally, typical Swedish softwood pellets (SW) that consist of a 50-50 mix between pine and spruce (*Pinus sylvestris* and *Picea abies*, respectively) were used as a reference fuel for a pelletized pure stemwood based wooden fuel. This enabled better comparability between the emission properties. The ash rich fuels, CH, BG, WH and RC, were mixed with SW on 50-50 mass ratio to improve the suitability and combustibility of the biomass fuels for cookstove combustion applications, including a better control of the ash content (i.e. certain inorganic elements). This was
done to enhance the stability of the combustion process and burning time.

The simplest cookstove used was the three-stone fire (3S); three bricks around an open fire to hold the cooking pot above fire. The rocket stove (RS) design is an improved stove model. However, the design is also fairly simple: the fuel sticks are inserted into the fuel shelf through the horizontal metal pipe and natural air draft causes a hot burning flame in the vertical
section, just below where the cooking pot is located during operation. The more advanced concepts included two pellet gasifier stoves where fuel pellets are loaded in a vertical pipe beneath the cooking pot and lit from above. During the operation, the pyrolysis front moves down on the fuel bed and releases volatile components. When the volatile gases reach the top of the cookstove, they are mixed with secondary air where they create a hot burning flame beneath the cooking pot. Although the operational principle was similar in both gasifier stoves, their air supply method differs in the following way:
the natural-draft gasifier stove (NDGS) uses natural air draft, while the forced-draft version (FDGS) is equipped with an adjustable fan to force primary and secondary air flows to allow a more efficient burn-out of soot and volatile organic matter.

Prior to experimentation, the fuels were pre-processed for the different types of cookstoves as follows: the fuel sticks used were chopped to pieces approximately 2 cm in diameter, 17 cm and 13 cm in length for the 3S and the RS, respectively, and
dried at room temperature. The pellet fuels were pelletized at the TEC-lab into dimensions of 8 mm x 15 mm (diameter x length, respectively). Fuel usage was thus standardized for the repeated WBT experiments according to cookstove type. The 3S and the RS were loaded with 100 g of stick fuel in the beginning of each WBT, and re-fueled slowly and continuously throughout the cooking simulation to maintain the flaming combustion. A typical number of re-fueling events was more than five. The gasifier stoves were loaded with 1 kg of pellets before lighting the fire, since they are batch-fired appliances where
fuel is not be added during operation.

**2.3 Ice nucleation experiments**

The instrument for the ice nucleation experiments was the SPIN (**SP**ectrometer for **I**ce **N**uclei), which is a commercial ice nucleus particle (INP) counter manufactured by Droplet Measurement Technologies Inc., Colorado, USA. The current





version of the instrument has been operational since January 2015, and it is presented by Garimella et al. (2016). The main

difference relative to the older version used by Ignatius et al. (2016) is improved temperature control.

Briefly, the SPIN is a continuous-flow diffusion chamber (CFDC) instrument with an IN-chamber design using parallel plate

geometry similar to the Portable Ice Nuclei Chamber PINC introduced by Chou et al. (2011) and the Zürich Ice Nuclei

Chamber ZINC described by Stetzer et al. (2008). The sample flow at 1 standard liter per minute (SLPM), is sandwiched

between two sheath flows of 4.5 SLPM each, and the residence time in the region where ice nucleation can take place is ca.

10 seconds. The diffusional flux of water vapor across the chamber is created via setting the ice-covered plates to different

sub-zero temperatures, and the relative humidity with respect to water and ice ($RH_w$ and $RH_i$, respectively) can be adjusted

through the temperature difference between the plates. The aerosol is exposed to an isothermal evaporation section, before

the detection with an optical particle counter (OPC). The temperature of the evaporation section was set to follow the

temperature of the aerosol sample in the growth section of the chamber. The temperature, $RH$:s and the path of the sample

flow, from here on referred as lamina flow, are modelled according to a 1D flow model by Rogers et al. (1988). The ice layer

was created to the IN chamber through cooling the walls to -32 $^O$C and filling the chamber with de-ionized water, which

resulted in a thin layer of ice on the walls of the chamber. After filling the IN chamber was purged of excess water and

vacuumed down to 70 mbar for 3 minutes, which removed loose ice and reduced roughness of the ice layer and thus the

background signal. Typically, the background signal was below 1 particle per liter in the beginning of the experiment and the

IN chamber was re-iced when it exceeded 10-15 particles per liter. This protocol allowed detection of the ice-activated

fraction from the order of $10^{-6}$ upwards, depending on the sample concentration.

A condensation particle counter (CPC, Airmodus model A20) was operated parallel to the SPIN for monitoring the sample

concentration and thus providing required information for calculation of the ice-activated fraction. The ice-activated fraction

$\alpha$ is therefore defined as

$$\alpha = \frac{N_{ice}}{N_{CPC}} \qquad (1)$$

where $N_{ice}$ is the background-corrected concentration of ice crystals detected by the OPC and $N_{CPC}$ is the concentration of

sample particles detected by the CPC. Background correction means that the frequency of background counts is linearly

interpolated between background checks and the corresponding temporal values are subtracted from the measured signal.

The studied ice nucleation modes follow the proposed definitions by Vali et al. (2015). Freezing conditions with $RH_w < 100$

% at the lamina flow are named as deposition mode and freezing above water saturation is referred to as immersion freezing,

because liquid droplet formation prior to freezing is expected. Condensation and immersion freezing modes are

indistinguishable in this instrument. The experiments were carried out using automated sequences that are available in this

version of the SPIN, to ensure comparability between experiments on different fuels and cookstoves.





The transient experiments were performed during the modified WBT using the transient sampling line, denoted A in Fig. 1. Fresh, polydisperse emission aerosol was introduced to the SPIN after desiccation to $RH_w < 5$ %. The concentrations were diluted to the order of $10^3$ cm$^{-3}$ to avoid vapor depletion inside the SPIN. The measurement sequence used in the transient

experiments was as follows: The lamina $T$ was fixed and $RH_i$ was increased via broadening the temperature difference between the IN-chamber plates. Eleven experiments were sampled transiently scanning at constant $T$ of -32 $^O$C and ramping the $RH_i$ from 100 % up to 160 %, which is close to the procedure used by Petters et al. (2009). This scan included studying both deposition and immersion modes because $RH_w$ ranged from approximately 73 % to 115 %. Two experiment scans where T was fixed at $T$ = -28 °C had similar $RH_i$ range and the only difference was in higher lamina temperature. The

experimental approach for the transient experiments is a standard operation procedure often applied in previous studies.

We introduced a different experimental approach with a focus on immersion freezing for the experiments involving sampling of aerosol from the chamber. This procedure is illustrated in Fig. 2, where a homogeneous freezing experiment on highly diluted ammonium sulfate (AS, dry mobility diameter of 350 nm) droplets is used as an example. All experiments were

carried out on similar automated ramps containing the following steps: First, the internal background was checked via sampling filtered dry air for 5 minutes before the lamina $T$ ramp began. As $T$ is descending, homogeneous freezing causes detection of ice crystals at about -37.9 $^O$C. When $T$ reaches the lowest set point of -43 $^O$C the background is checked again via filtering sample flow at the inlet for 3 minutes, before opening it again and scanning the $T$ back to -32 $^O$C (ascending ramp). The phase change back to droplets is detected at -38.9 $^O$C, when the ice-activated fraction decreases to below $10^{-3}$.

The difference in ice onset and offset temperatures is most likely caused by cold pockets that may occur in the IN chamber during cooling in descending ramps. When the temperature control uncertainty is defined as one standard deviation from the averaged lamina temperature, it was observed that the temperatures typically deviate 0.9-1.0 $^O$C and 0.2-0.3 $^O$C from the set value of the lamina temperature (see the shaded area around lamina temperature graph in Fig. 2) in descending and ascending ramps, respectively. Therefore, all data obtained from the descending ramps were omitted from the analysis and

only ascending ramps were studied. The total sampling time of about 2 hours, which was used in chamber experiments, enabled up to three repetitions for each WBT emission sample. This sampling time allowed 2-3 upward $T$ scans for each chamber experiment, and the reproducibility of the experiments will be discussed in the Section 3.4.

Separation between liquid droplets and ice crystals was carried out using a basic particle size-threshold method in the

immersion mode experiments. Based on the homogeneous freezing experiments and all the other experiments we carried out under these conditions, the liquid droplets do not grow larger than approximately 4.5 μm in the SPIN when the sheath and sample flows of 9 LPM and 1 LPM, respectively, are used and the particle residence time in the IN chamber is 10 seconds. This can be seen in Fig. 2 between 20.38-20.49 hrs and 20.75-20.87 hrs when the inlet is open, and the lamina $T$ is above homogeneous freezing temperature. Therefore, particles larger than 6 μm are considered ice crystals in all these experiments.

The approach with operating SPIN with potential co-existence of droplets and ice crystals is unusual, and it has to our

knowledge previously only been applied for investigating the homogeneous freezing temperature (e.g. Ignatius et al., 2016). In this context, it should be noted that the evaporation section is less efficient for SPIN compared to some other CFDCs. The SPIN OPC was designed to detect for particle depolarization to be used for discrimination of droplets versus ice crystals on a particle-by-particle basis. However, we found that the depolarization data obtained with the SPIN5 OPC during this

campaign were not of a quality sufficient for such an approach. We never observed any droplets with optical diameters larger than 6 μm in any experiment with the approach described above, so we found it meaningful to infer an ice-active fraction by use of that size threshold, since the ice crystal mode typically was centered around larger sizes. The potential biases in the ice-active fraction associated with this approach are discussed in more detail in a later section. One advantage associated with this approach was that we observed formation of a significant droplet mode for all experiments, which clearly indicated

that the supersaturation sufficed to produce cloud droplets in order to investigate immersion freezing – even for particles with a low CCN activity.

It has previously been shown that the CFDC ice concentrations often are biased low by a factor of 3 and potentially up to a factor of 10 depending on the operation conditions [DeMott et al., 2015; DeMott et al., 2017; Garimella et al., 2017]. This

bias is due to non-ideal behaviour of instruments in real life, when the sample and sheath flows cannot follow the theoretical streamlines ideally. Hence, we will also estimate this effect and its possible effect on the results of this study.

## 3 Results and discussion

We present the IN results from transient and chamber experiments, compared to homogeneous freezing experiments whenever the experimentation included sample temperatures near or below homogeneous freezing temperature from the

droplet freezing test. Ice onset is defined as an ice-activated fraction of $10^{-3}$. A result is considered positive if the ice onset has taken place under conditions when the sample temperature or $RH_i$ have been significantly higher or lower, respectively, than conditions required for homogeneous freezing. The result is considered negative if ice crystals have not been observed or if there is no distinguishable difference to homogeneous freezing conditions.

### 3.1 Transient experiments

Transient experiments are summarized in Table 1 and they all were conducted on fresh, polydisperse samples whose size distributions were measured simultaneously during the WBTs. All samples were dominated by ultrafine particles by number, and the fraction of particles larger than 250 nm was typically below 5 % of the total particle number concentration in all experiments. It can be expected that the size distributions and emission properties varied more on the 3S and the RS than on gasifier stoves due to re-fueling, i.e. addition of fuel sticks during the simmering phase. Regarding that the concentration

introduced to the SPIN was typically 1000-2500 cm$^{-3}$ in transient experiments, ice nucleation was detectable from the order of $10^{-6}$ upwards and larger accumulation mode particles accounted for concentrations up to about 100 cm$^{-3}$. Ice crystal



formation was not observed within the detection limit in any experiment at $T$ = -32 $^{O}$C or above. These 11 observations indicate that the fresh aerosol emission from these four stoves are an active ice nucleators in neither deposition nor immersion mode at the given experiment temperatures. The emissions from all experiments were dominated by ultrafine particles by number, which clearly did not contribute to heterogeneous ice nucleation for the given temperatures.

## 3.2 Chamber experiments

The chamber experiments served two major purposes, (i) it allowed for studying aerosol emissions representing an average over several combustion phases, and (ii) it allowed for focused ice nucleation studies of the soot particles. Regarding that typical aerosol injection time was up to 40 minutes, the aerosol emissions from the 3S and the RS stoves are highly variable depending on the combustion phase e.g. fuel addition, flaming and smoldering. In contrast, the aerosol emissions from the gasifier stoves were constant during most of a given transient experiment except for the ignition and the extinction phases, which allowed for SPIN scans over almost constant aerosol over time. Most aerosol injections into the aerosol storage chamber were carried out over roughly a full combustion cycle representing the different combustion phases. Typical particle mass concentrations in the storage chamber was at the order of 10-100 $\mu g\,m^{-3}$, and the $RH_w$ was typically at the order of 20-25%.

The immersion mode was studied on ramps with fixed $RH_w$ = 115%, and the chamber experiments are listed in Table 2. The FDGS was modified in two experiments to simulate usage of a poorly functioning stove, and the experiment was repeated twice on SW fuel. This simulation was done by reducing the secondary air supply, which resulted in more inefficient and incomplete combustion and increased the production of large soot particles significantly. The "*pot height*" experiments were done using different, less favorable heights of the cooking pot. The three such experiments were done on designed operation conditions of the FDGS and SW pellets, but the cooking pot was intentionally lifted to achieve production of larger particles: the real-time size distribution was monitored on the FPA throughout the experiment and the pot height was adjusted in a way that an increased production of large soot particles was observed. In typical cases, an offset of 8-10 cm above designated pot height affected production of large soot particles significantly.

The particle number size distributions were measured using the SMPS from the storage chamber, and therefore the sample sizes, as Fig. 3 shows, are based on dry particle mobility. The size distributions were by number typically highly dominated by ultrafine and largely inorganic hygroscopic particles as indicated by CCN measurements (although not presented in this study). All ice nucleation experiments involving poly-disperse aerosol indicated that the ultrafine particles did not play a role as INPs for the investigated conditions. The chamber experiments focused on quasi-monodisperse size selected soot particles. We aimed for sampling the largest possible particle mobility diameter (200-500 nm) with reasonable number concentrations (50-200 cm$^{-3}$) resulting in minor contributions of multiply charged particles for any given experiment. The presented size distributions represent the situation when an activated fraction of 10$^{-3}$ was observed in the SPIN in each


shown cookstove and fuel combination. Particles larger than 250 nm accounted for up to 15 % of the total number concentration in all chamber experiments, which indicates that soot emissions were present in all studied cases.

The IN efficiencies of the emissions are presented in the following paragraphs. All experiment results are presented when $10^{-3}$ activated fraction was observed during the first ascending $T$ ramp, when the particle storage period in the chamber has been

close to one hour. The bias regarding activated fraction due to restrictions in real-life CFDC instruments is being discussed separately in Chapter 3.4, along with reproducibility of the observations.

Starting from the simplest cookstove concept, the 3S produced $10^{-3}$ ice-activation between -38.1 $^{O}$C and -38.3 $^{O}$C. Although these values are close to the observed homogeneous freezing temperature, comparison to a similar experiment on diluted AS

droplets reveals that ice onset due to homogeneous freezing occurs at -38.9 $^{O}$C with this operation of the SPIN. The observed homogeneous freezing with $10^{-3}$ ice activation at an average lamina temperature of -38.9 $^{O}$C may occur at slightly lower temperature than expected. It is not clear at which temperature homogeneous freezing in SPIN can be expected due to the sample aerosol being exposed to a span in temperature and RH, and uncertainties related to droplet sizes and residence times above the evaporation section as discussed by Ignatius et al. (2016). Furthermore, the ice-active fraction is in the current

study biased low for two reasons: (i) only a fraction of the sample aerosol is focused in the lamina (Garimella et al., 2017), and (ii) only the larger size fraction of the ice crystals (optical diameter >6 μm) is included in the ice-active fraction (Fig. 2). We estimate that correction for those biases in the homogeneous freezing experiments will result in $10^{-3}$ ice activation between -38.5 $^{O}$C and -38.0 $^{O}$C in average lamina temperature. We ascribe this relatively lower and reproducible homogeneous freezing temperature for the current version of SPIN to highly improved temperature control relative to earlier

versions of SPIN (Ignatius et al., 2016). Potential biases and errors are discussed in more detail later.

Figure 4 shows the results on two fuels, CAS and SES combusted with the 3S, and three different sample sizes. The errors related to the average temperature of the lamina are presented above as the standard deviation based on the uppermost 13 pairs of thermocouples on the warm and cold plates, respectively, as was done by Garimella et al. (2016). The uncertainty

analysis is based on modelled lamina temperature, the shaded area represents one standard deviation of lamina $T$ from averaged lamina conditions at the time of each observation, when the 13 topmost thermocouple pairs are considered individually. The uncertainty analysis confirms that there is a significant difference in $10^{-3}$ onset temperatures between the 3S experiment emissions and the homogeneous freezing experiment. High experiment reproducibility that will be discussed in Section 3.4 indicates very good agreement between repeated ramps and thus strengthens the validity of these observations.

Size dependency that could be distinguished from the instrument uncertainty between 250, 300 and 500 nm particles was not observed, which is likely due to differences in chemical composition between the individual experiments.





The RS emissions were slightly more active INPs than ones from the 3S, the $10^{-3}$ activated fraction was achieved between -37.7 $^\circ$C and -37.9 $^\circ$C which is 1.0-1.2 $^\circ$C higher than observed homogeneous freezing in the AS experiment, as can be seen

from Table 3. Indications of size-dependency was also not observed in these experiments. The results from regular gasifier stove experiments are presented in Fig. 5, and conditions for $10^{-3}$ ice-activated fraction were reached at -37.8 $^\circ$C on the FDGS with CH+SW pellets, and at -38.1 $^\circ$C on the NDGS with SW pellets during standard operation conditions. Once again, a clear size-dependency was not observed, the 260 nm particles from the experiment on the FDGS appear to be slightly more IN active than 400 nm particles from the NDGS. When compared to standard tests on the 3S and the RS, these

observations consistently show that the experimented soot particles activate the heterogeneous ice nucleation in warmer temperatures than where homogeneous freezing starts.

In a few selected experiments, we investigated how modified combustion conditions possibly would influence the ice activity of the emitted soot particles. Five different modified experiments with the FDGS and SW pellets were used to inject

aerosol particles into the chamber from which SPIN sampled size selected soot particles. Two of them were carried out with the secondary air supply being intentionally blocked, and the pot height was adjusted to increase the particle size in three experiments. The ice active fractions versus sample temperatures are presented in Fig. 6. For two of the samples, heterogeneous ice activity was observed at temperatures about 4 $^\circ$C (blocked secondary air supply) and 6 $^\circ$C (elevated pot height) higher than observed for homogeneous freezing and an ice active fraction of $10^{-3}$, respectively. These experiments

were repeated, and the ice activity was comparable to homogeneous freezing for other intendedly identical experiments. This indicates that very minor changes in combustion conditions significantly influence the ice nucleating ability of freshly produced soot particles.

It is still largely an open question, which soot particle parameters influence the ice nucleating ability. However, the extensive

online characterization of the aerosol applied in this work allows us to investigate such potential links. In Fig. 7, we present 6 different physico-chemical particle properties for the aerosol samples included in Fig. 6. The CCN activity presented as the apparent $\kappa$ ($\kappa_a$) and the effective density ($\rho_e$) were inferred for particles with a mobility diameter of 350 nm, and the results are presented in the two upper panels of Fig. 8. Information about optical properties (the ratio between the absorption at a wavelength of 370 nm to 880 nm), relative abundance of refractory oxygen species ($C_3O_2^+/C_3^+$), the ratio of organic aerosol

mass (Org) (in the $m/z$ range 13-330 Th) to rBC mass, and indirect information about the nanostructure ($C_{mid}/C_{all}$) of the poly-disperse aerosol is presented in the lower panels in Fig. 7, with the three latter properties being inferred from the SP-AMS data.

The $\kappa_a$ was inferred as described by Petters and Kreidenweis (2007) and the $\rho_e$ was inferred in a similar fashion as done by

Rissler et al. (2013). The $\kappa_a$ values are very low for all the samples with only minor variations, and the $\rho_e$ values are also low for all the samples with minor variations. These values of $\rho_e$ (0.2-0.3) are typical for fractal-like rBC dominated



agglomerates from several different sources (Rissler et al. 2013). We observed a tendency towards decreasing $\kappa_a$ and $\rho_e$ with increasing mobility diameter in the soot mode throughout the campaign, so the presented values can be considered upper estimates of what may be relevant for the larger quasi-monodisperse soot particles studied with SPIN.


It is noteworthy that cases B500 and B450 showed elevated absorption in the UV region, despite the low OA to BC ratios. These two cases were also associated with an increased fraction of carbon fragments in the mid-carbon range $C_6^+$ to $C_{29}^+$ in the refractory carbon mass spectrum detected upon laser vaporization with the SP-AMS. Malmborg et al. (2019) and Török et al. (2018) linked increased UV absorption to the occurrence of large carbon fragments in the SP-AMS mass spectrum,

decreased fringe length detected by HR-TEM and refractory (partly pyrolyzing) organic carbon in thermal-optical analysis for emissions from the Mini-CAST flame soot generator. These findings suggest that such relationships are also present for refractory carbon dominated particles from cookstoves. Particularly, experiments B500 and B450 were carried out by blocking the secondary air inlets in the FDGS, which facilitated emissions of refractory carbon species with elevated UV absorption. The ratio of $C_3O_2^+/C_3^+$ showed only small variations between experiments (0.005-0.007) suggesting that the

occurrence of refractory oxygen species (for example surface oxides) did not vary much between experiments.

Our detailed aerosol characterization presented in Fig. 7 indicates that we did produce particles with slightly different properties for intendedly identical experiments, as also was indicated by the observed difference in the ice nucleating ability. The low hygroscopicity, the low effective density and the optical properties indicate that the studied soot particles were not

covered or coated by significant amounts of soluble material despite the presence of some organic compounds. It is noteworthy that the two more ice active soot samples (B450 and K400#1) are significantly different with regards to several properties. The only minor trend correlating with ice-activity is a slightly lower ratio of refractory oxygen species $(C_3O_2^+/C_3^+)$ for the two more ice active soot samples relative to the less ice active samples. However, it is questionable whether that trend is significant. In general, these supportive data indicate that either (i) the inferred properties are not

determining the ice nucleating ability, or (ii) a potential for complex combinations of different soot particle properties being of relevance for the ice nucleating ability. Further studies are needed in order to reach any firm conclusion in this matter.

**3.3 Experiment reproducibility and bias in ice crystal detection**

The typical sampling time was approximately 2 hours in all chamber experiments, and it enabled 2-3 ramp pairs and thus the corresponding number of repeated ascending ramps for each experiment on the SPIN. In this section, we discuss the

reproducibility of the observations to validate the results and to evaluate the measurement precision of the SPIN. The results from repeated ramps are summarized in Table 3 where we present the ice onset temperatures and their uncertainty at the lamina $T$. Observation numbers #1, #2 and #3 represent the IN onset after chamber residence times of approximately 20, 40 and 60 minutes, respectively, after filling the chamber was completed in each WBT. These comparisons show that the results from repeated ramps are generally in good agreement with each other, and the differences in onset temperature remain within



the typical deviation of the SPIN (see Chapter 2.3). It can be expected that the most prominent chamber effect has been coagulation of the ultrafine particles to large soot particles, but this has not affected to ice nucleating ability.

The errors related to the average temperature of the lamina are presented above as the standard deviation based on the uppermost 13 pairs of thermocouples on the warm and cold plates, respectively. It should also be mentioned that across the
width of the lamina, there is an additional span in temperature typically at the order of ±0.4°C, so a fraction of the aerosol will be exposed to further lower or higher temperatures, respectively, compared to just the variability in average lamina temperature as presented in e.g. Fig. 2. However, we find that the variability in average lamina temperature is a reasonable estimate of the error in sample temperature for the experiments presented in Figures 4-6 considering the high reproducibility of the results presented in Table 3.


The presented ice active fractions are biased low for two main reasons as mentioned above. As it can be seen from Figures 4-6, the ice active fractions for a temperature of -41°C is typically at the order of 0.1 to 0.2, while we would expect an ice active fraction close to 1 for homogeneous freezing at that temperature below -40°C. These observations confirm that the presented ice active fractions are biased low. Previous studies show that only a fraction of the sample aerosol is focused in
the lamina. DeMott et al. (2015) found that the ice active fraction should be upscaled by a correction factor of 3 for a CFDC and somewhat similar operation conditions to what we apply in our study. Garimella et al. (2017) reported upscaling factors for SPIN in the range from 1.5 to 10 depending on the operation conditions due to only a fraction of the sample aerosol being focused in the lamina.

In the current study, we have droplets and ice crystals coexisting after the evaporation section for a significant range of conditions, and with our established ice crystal threshold (optical diameter >6 μm), only a fraction of the ice crystal mode is included in our reported ice active fractions. This underestimation depends on the lamina temperature. Investigation of the depolarization ratio of particles with an optical diameter in the sub-micrometer range with the SPIN OPC confirms that ice dominates in this size range at the lowest temperatures <-40°C. Hence, the ice active fraction inferred from our procedure is
underestimated, which is likely to be significantly more pronounced in the lowest temperature as can be seen from Fig. 2. Furthermore, the ice active fraction may also be underestimated due to (i) losses of ice crystals immediately above the OPC, which to our knowledge has not been studied in detail so far for SPIN, and (ii) a relatively small fraction of particles activating into droplets inside SPIN – leading to a reduced fraction for detection in the immersion mode. The potential losses have to our knowledge not yet been characterized in detail – while the ice-active fractions observed for homogeneous
freezing of dilute ammonium sulfate droplets and droplets formed on hydrophobic particles typically resulted in very similar ice-active fractions (e.g. Fig. 6).


The considerations discussed above lead us to estimate a likely upscaling factor at the order of ~3-5 of the ice active fraction for temperatures around -33$^O$C, while the upscaling factor is likely to approach 5-10 for temperatures in a neighborhood of -40$^O$C. Further studies are needed to assess these biases in more detail – and their relative importance for different operation conditions of the CFDC instrument.

**4 Summary and conclusions**

The SUSTAINE experiment campaign provided an excellent opportunity to study the IN efficiencies of emissions from different biomass-fired cookstove designs under well-controlled laboratory conditions, which enabled comparability between individual experiments. Two cookstoves, the 3S and RS, represented designs that are commonly used in daily household cooking in sub-Saharan Africa. The two more sophisticated designs, the NDGS and the FDGS, are currently under research for sustainable development and their popularity can be expected to increase along with economic development in this and similar regions. This study shows that even small changes in combustion conditions can significantly affect the IN abilities of emission particles. The regional daily usage by at least 500 million people suggests that the solid biomass burning emissions from cookstoves are a significant source of atmospheric particulate matter in sub-Saharan Africa. The CFDC instrument in this study, the SPIN, showed relatively good performance and high reproducibility of experiments.

We conclude on the experiments that the fresh, polydisperse emissions from cookstoves have a low INP potential at experimented temperatures (-28 $^O$C, -32 $^O$C). All emissions were heavily dominated by ultrafine particles that clearly showed poor INP activity. Moreover, their residence time in the atmosphere is relatively short due to deposition, coagulation and external mixing with other atmospheric particle species. Accumulation mode particles that were present in the transient experiments were not observed to activate heterogeneous ice nucleation at -32 $^O$C in neither freezing mode, from which it can be concluded that the studied cookstove emissions have low IN activity at that temperature also in the immersion mode freezing. The results from chamber experiments on accumulation mode soot particles show that they can induce heterogeneous ice nucleation in higher temperatures than what is required for homogeneous freezing, as happened in most of the experiments. Therefore, we conclude that usage of the cookstoves can emit potential INPs in the atmosphere, and thus affect cloud properties. The chamber experiments also support the outcome of transient experiments that included the same cookstove-fuel combinations, because the experimental procedures included sampling at corresponding $RH_w$ of 115% but in lower temperatures. With these observations combined, we conclude that the fresh cookstove emissions that were tested both transiently and from the chamber do not contain components that are active INPs at or above a temperature of -32 $^O$C. All but two chamber experiments show an obvious difference to homogeneous freezing experiment, and therefore these emissions may well be of relevance in their effect on atmospheric radiation budget. Their atmospheric importance may be of second order in comparison to e.g. mineral dusts as Vergara-Temprado et al. (2018) conclude in their study, but daily usage by hundreds of millions of people can still make cookstove emissions a significant regional source of atmospheric INPs. Despite of the chamber experiments consistently showing IN activity above homogeneous freezing temperature, it still needs

to be noted that the studied emissions were relatively fresh: atmospheric ageing processes can affect the IN properties before the emissions reach the upper atmosphere, as is being supported by Häusler at al. (2018). The effect of atmospheric ageing can be studied via using e.g. oxidation reactors in future studies.

Deterioration of combustion efficiency was observed to increase the INP potential of the emission particles, which is likely due to elevated large soot particle production in incomplete combustion of the tested biomass fuels. This, when combined with results from transient experiments and ones with standard combustion conditions, indicates that even modest changes in combustion efficiency can drastically alter the ice-nucleating capabilities of the emissions. Our analysis on physico-chemical properties of the emissions revealed slight differences in studied properties of soot particles that were present in the most ice-

active results, yet these properties cannot define the IN efficiency alone. Another possibility is that the IN efficiency of soot is potentially determined by a complex combination of multiple particle properties. We conclude this section that the inferred physico-chemical properties cannot determine the ice-nucleating ability of soot particles in this study. We recommend further studies for finding the relation between soot nanostructure and its potential as INPs.

It is worth emphasizing that all experiments of this study were carried out in well-controlled laboratory conditions and using standardized test procedures. Contrary to that, real-life use can differ significantly from these experiments in many factors, such as in fuel preprocessing, fuel properties, technical stove conditions and practical cooking procedures. These all affect the combustion and emission performance that can have a prominent effect on ice-nucleating abilities of emitted aerosol particles, which this study shows. Therefore, the relevance between these observations and real-life use should be explored

before the contribution of biomass-burning emissions can be further evaluated in global perspective.

**Data availability**

The data set is available upon request from Kimmo Korhonen (kimmo.korhonen@uef.fi).

**Competing interests**

The authors declare that they have no conflict of interest.



**Author contribution**

TBK planned the ice-nucleation experiments on the SPIN instrument that was operated by KK during the campaign. AV, JF, KK, MK and TBK participated in data analysis of ice-nucleation experiments and/or interpretation of results. RL and RLC prepared the experimental set-up for water boiling tests and operated the combustion facility during the campaign. AE, CA, JP, TBK and VB-M participated in collection of supportive data and/or interpretation of results. BS, CB and JP participated as the organizers and supervisors of the SUSTAINE experiment campaign. AV and KEJL achieved funding for the SPIN
instrument. All authors participated in scientific discussions on this study and reviewed/edited the manuscript during its preparation process. KK prepared the manuscript with contributions from all co-authors.

**Acknowledgements**

This work was financially supported by the Swedish Research Council FORMAS through the project Sustainable Biomass Utilization in Sub-Saharan Africa for an Improved Environment and Health (Dnr. 942-2015-1385) and Atmospheric cloud droplet formation and ice formation of wood combustion aerosols (2015-992). T.B. Kristensen gratefully acknowledges funding from the Swedish Research Council (VR) grant no. 2017-05016. LU-EAT researchers acknowledge financial support from the Swedish Research Councils VR (projects 2018-04200 and 2013-05021) and FORMAS (project 2013-
01023). University of Eastern Finland and Finnish Meteorological Institute acknowledge Academy of Finland, Centre of Excellence (grant no. 272041) and North-Savo Council - European Regional Development Fund's (project no. A32350). Ricardo Carvalho acknowledges the Postdoctoral grant JCK-1516 funded by the Kempe Foundation.

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






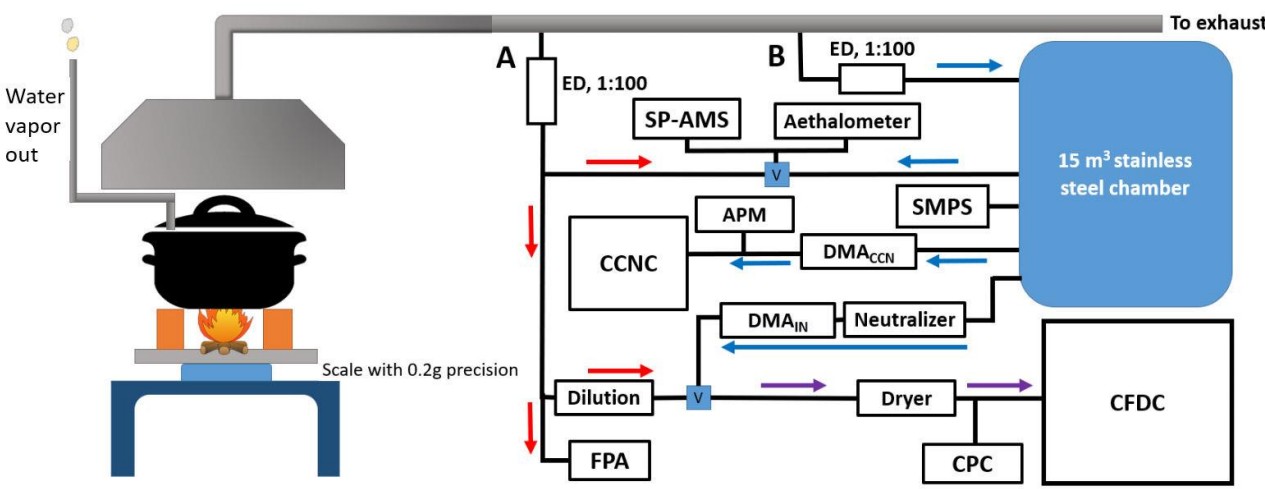

**Figure 1: Schematic of the experimental set-up used to conduct the WBT modified version relevant to the IN experiments. Sample lines A and B were used for transient and chamber experiments, respectively. Blue squares with letter "V" indicate valves which were used for switching between the sampling lines. The red and blue arrows show directions of sample flow related to transient and chamber experiments, respectively. The purple arrows show the flow direction used in both types of experiments, depending**

**on position of the relevant switch valve. The acronyms are defined as follows: APM = aerosol particle mass analyzer, CCNC = cloud condensation nuclei counter, CFDC = continuous flow diffusion chamber ice nucleus counter, CPC = condensation particle counter, DMA = differential mobility analyzer, ED = ejector dilution, FPA = fast particle analyzer, SMPS = scanning mobility particle sizer, SP-AMS = soot particle aerosol mass spectrometer. The dimensions and sample line lengths are not in scale to each other.**





**Figure 2: The immersion mode experiment procedure example from homogeneous freezing experiment with dilute 350 nm ammonium sulfate seeds. Upper panel: Symbols $T_w$, $T_c$, and $T_a$ represent average temperatures for warm wall, cold wall and sample aerosol (lamina), respectively. The shaded black area in the top graph represents one standard deviation from the averaged lamina temperature. Lower panel: Events marked "BGD" depict background signal checks. The color bar indicates the logarithmic intensity of particle counts and the dashed black line the ice threshold size of 6 µm. The gap in detection near 3 µm is a specific artefact in the OPC in this SPIN unit.**

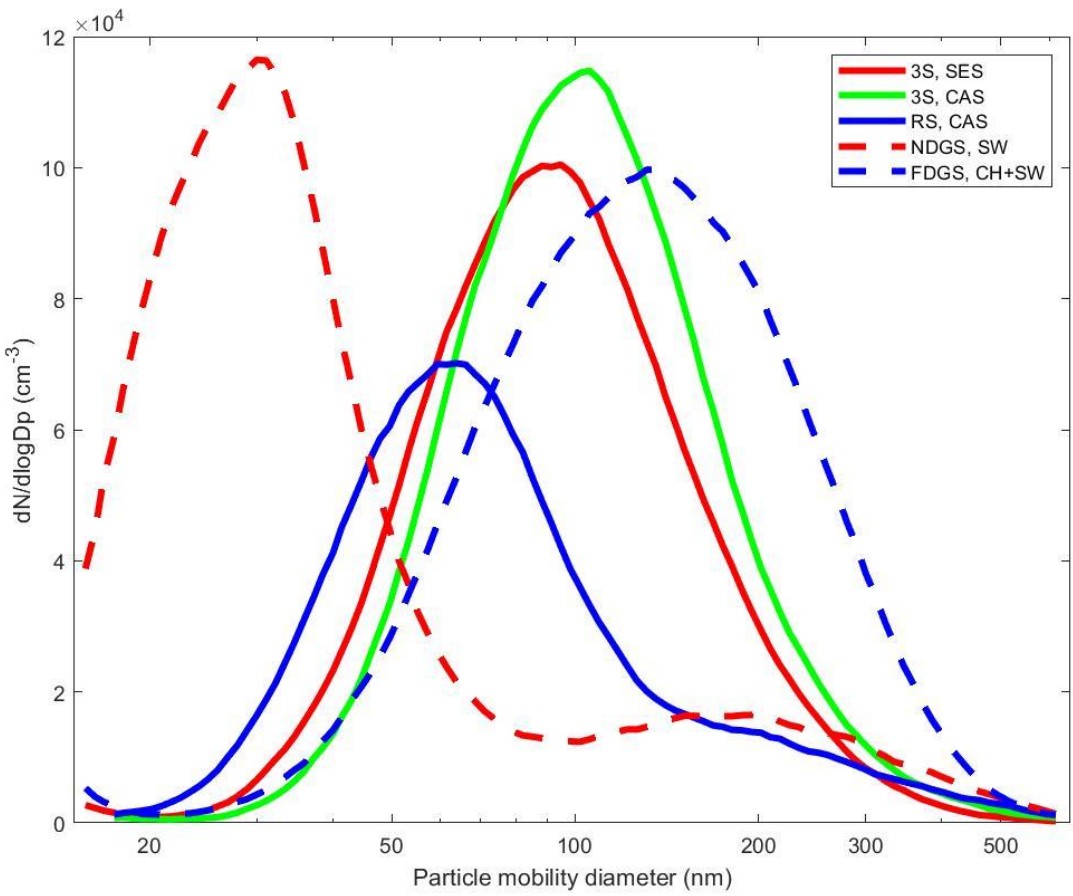


**Figure 3: Examples of particle mobility size distributions from chamber experiments on different cookstoves, during the time when an activated fraction of $10^{-3}$ was observed in the SPIN.**





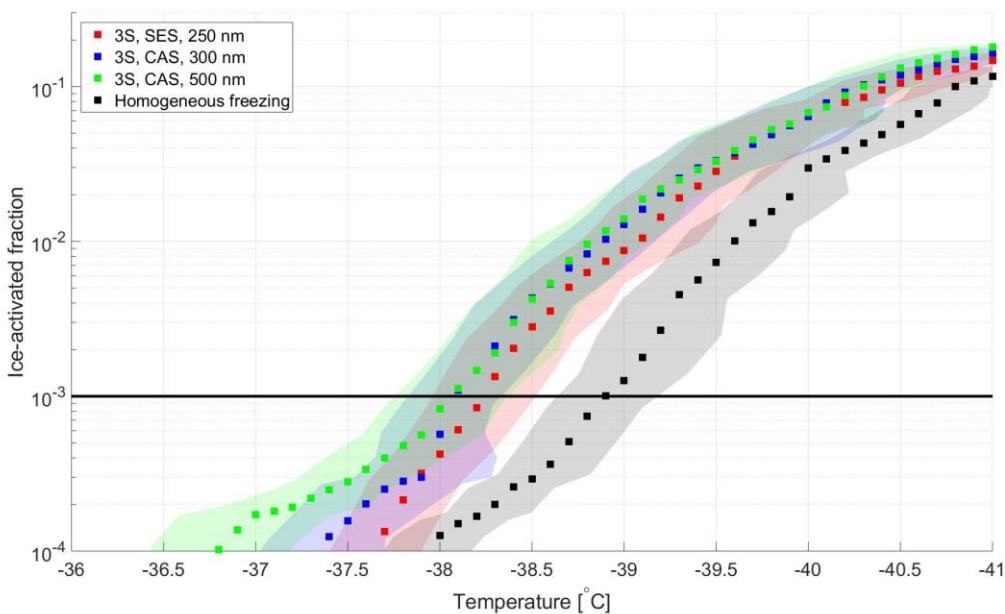

**Figure 4: Ice-activation spectra of emissions from the 3-stone fire at $RH_w$ = 115%, chamber experiments. Each shaded area of respective color represents ± one standard deviation on lamina temperature during each observation. The solid black line presents the $10^{-3}$ activation threshold. The ice-activation spectrum for homogeneous freezing is included for comparison.**




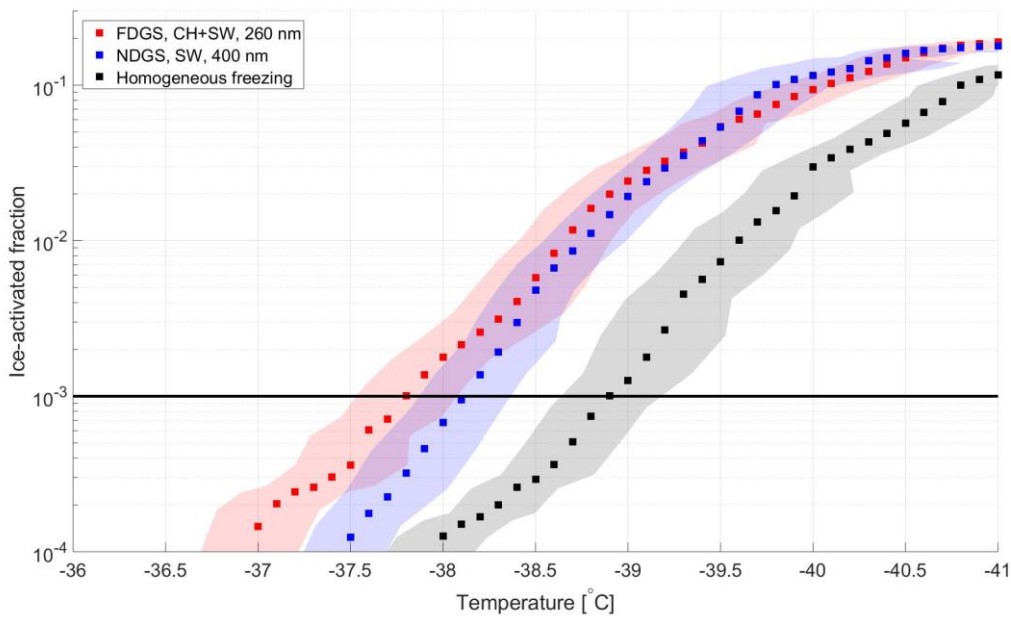

**Figure 5: Ice-activation spectra of emissions from gasifier stoves at $RH_w$ = 115% at regular operation conditions. Each shaded area of respectful color represents ± one standard deviation on lamina temperature during each observation. The solid black line presents the $10^{-3}$ activation. The ice-activation spectrum for homogeneous freezing is included for comparison.**



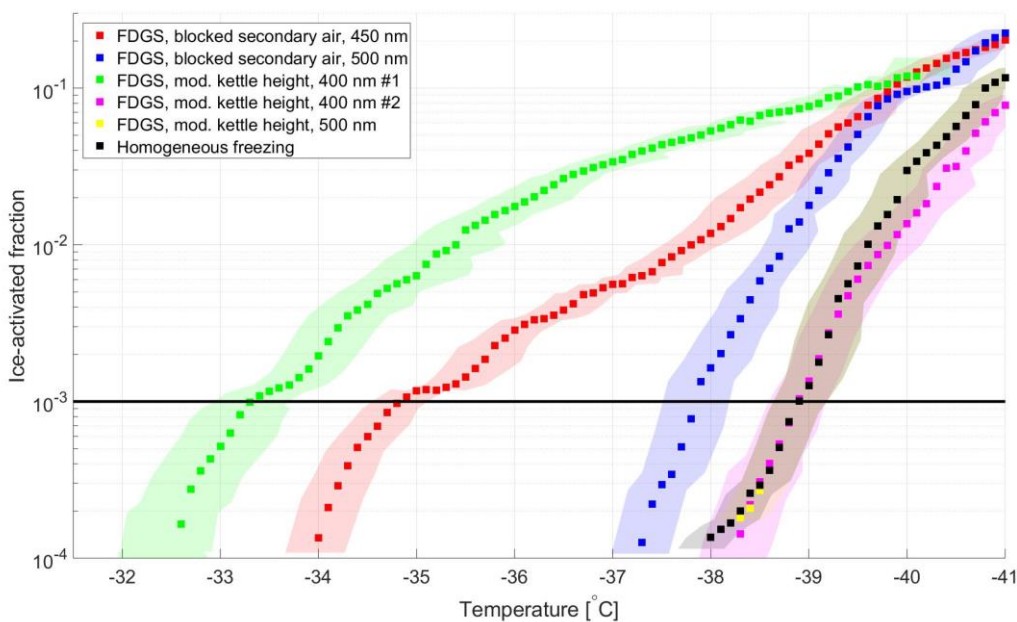

**Figure 6: Ice-activation spectra of emissions from combustion of SW pellets in forced-draft gasifier stove, with modified combustion conditions at $RH_w$ = 115%. Each shaded area of respectful color represents ± one standard deviation on lamina temperature during each observation. The solid black line presents the $10^{-3}$ activation. The ice-activation spectrum for homogeneous freezing is included for comparison.**



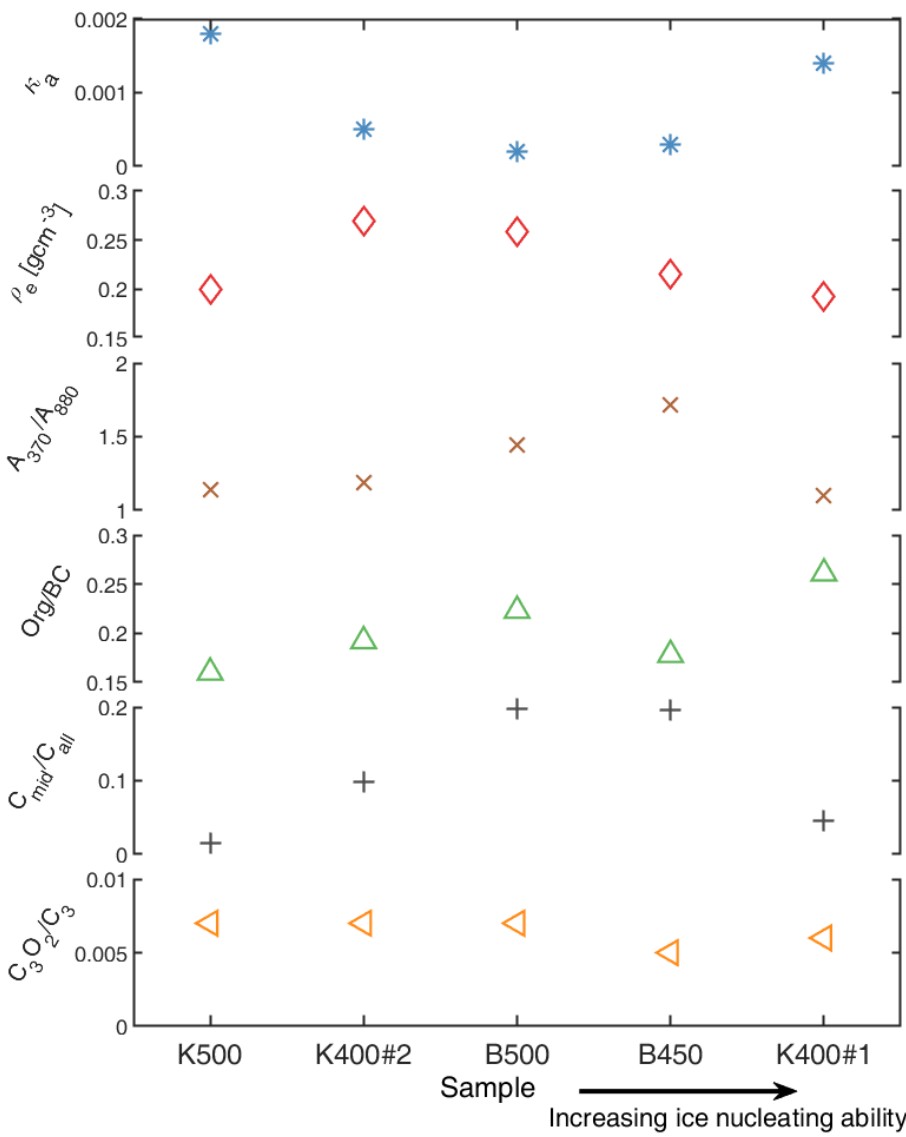


**Figure 7: Physico-chemical properties of soot particles in special experiments where the combustion conditions of sw in the FDGS were modified. The five aerosol samples presented have been re-ordered in a way so that the samples with higher ice nucleating ability are located further to the right. The properties from the top to the bottom represent the CCN activity ($\kappa_a$), the effective density ($\rho_e$), the ratio between the absorption at a wavelength of 370 nm versus 880 nm ($A_{370}/A_{880}$), the ratio of organic aerosol mass to refractive BC mass (Org/BC), indirect qualitative information about the nanostructure ($C_{mid}/C_{all}$) and the relative abundance of refractory oxygen species ($C_3O_2^+/C_3^+$), The two uppermost panels are for quasi-monodisperse soot particles with a mobility diameter of 350 nm, while the other properties are inferred for polydisperse aerosol. Letters K ja B refer to modified cooking pot height and blocked secondary air experiments, respectively, in labels of the x-axis.**







**Table 1: Summary of transient experiments on polydisperse sample particles on $RH_i$ scan 100-160 %, and the ice crystal detection limit. The detection limit depends on sample concentration and is defined as the lowest activated fraction when detection of ice crystals is distinguishable from the background signal in data averaged over 10-second periods.**

| Cookstove | Fuel | Temperature | Detection limit |
|---|---|---|---|
| 3-stone fire | BIR | -32 °C | $6.6 \times 10^{-6}$ |
| Rocket stove | CAS | -32 °C | $7.3 \times 10^{-6}$ |
| Rocket stove | CAS | -32 °C | $4.0 \times 10^{-6}$ |
| ND gasifier | SW | -32 °C | $5.0 \times 10^{-6}$ |
| ND gasifier | CH+SW | -32 °C | $1.3 \times 10^{-6}$ |
| ND gasifier | BG+SW | -28 °C | $5.0 \times 10^{-6}$ |
| ND gasifier | BG+SW | -32 °C | $3.4 \times 10^{-6}$ |
| FD gasifier | SW | -32 °C | $1.2 \times 10^{-5}$ |
| FD gasifier | BG+SW | -32 °C | $6.2 \times 10^{-6}$ |
| FD gasifier | WH+SW | -32 °C | $3.9 \times 10^{-6}$ |
| FD gasifier | RC+SW | -32 °C | $4.4 \times 10^{-6}$ |






**Table 2: Summary of chamber experiments. The "*pot height*" experiments represent modified combustion conditions when the cooking pot was intentionally placed above the designated height of the respective cookstove.**

| Cookstove | Fuel | Sample size (nm) |
|---|---|---|
| 3-stone fire | SES | 250 |
| 3-stone fire | CAS | 300 |
| 3-stone fire | CAS | 500 |
| Rocket stove | CAS | 250 |
| Rocket stove | CAS | 350 |
| Rocket stove | CAS | 450 |
| ND gasifier | SW | 400 |
| FD gasifier | CH+SW | 260 |
| Modified FD gasifier | SW | 450 |
| Modified FD gasifier | SW | 500 |
| FD gasifier, pot height #1 | SW | 400 |
| FD gasifier, pot height #2 | SW | 400 |
| FD gasifier, pot height #3 | SW | 500 |






**Table 3: Ice onset ($10^{-3}$ activation) temperatures when ascending temperature ramps were repeated during each chamber experiment, calculated from background-corrected signal. The last entry shows results from repeated ramps in homogeneous freezing test. The uncertainties equal to ± one standard deviation in the lamina temperature at the moment of detection. The "*pot height*" experiments represent modified combustion conditions when the cooking pot was intentionally placed above the designated height of the respective cookstove.**

| Cookstove | Fuel | Sample size (nm) | Ice onset $T$ [$^O$C], ramp #1 | Ice onset $T$ [$^O$C], ramp #2 | Ice onset $T$ [$^O$C], ramp #3 |
|---|---|---|---|---|---|
| 3-stone fire | SES | 250 | -38.3±0.2 | -38.2±0.2 | -38.2±0.4 |
| 3-stone fire | CAS | 300 | -38.3±0.5 | -38.2±0.3 | -38.6±0.4 |
| 3-stone fire | CAS | 500 | -37.8±0.3 | -38.0±0.2 | -38.1±0.3 |
| Rocket stove | CAS | 250 | -38.2±0.3 | -38.1±0.3 | - |
| Rocket stove | CAS | 350 | -37.8±0.3 | -38.1±0.3 | - |
| Rocket stove | CAS | 450 | -37.9±0.4 | -37.9±0.3 | - |
| ND gasifier | SW | 400 | -38.2±0.3 | -38.2±0.3 | - |
| FD gasifier | CH+SW | 260 | -37.8±0.3 | -38.1±0.3 | - |
| Modified FD gasifier | SW | 450 | -34.7±0.3 | -35.1±0.4 | -35.1±0.4 |
| Modified FD gasifier | SW | 500 | -38.1±0.3 | - | - |
| FD gasifier, pot height #1 | SW | 400 | -33.4±0.3 | -33.9±0.3 | -34.0±0.3 |
| FD gasifier, pot height #2 | SW | 400 | -38.9±0.4 | -38.9±0.4 | - |
| FD gasifier, pot height #3 | SW | 500 | -39.3±0.4 | - | - |
| - | AS | 350 | -38.9±0.3 | -39.1±0.3 | -38.9±0.3 |