# Peer review of "Ice nucleating ability of particulate emissions from solid biomassfired cookstoves: an experimental study"

_Atmospheric Chemistry and Physics, 2019_

## Referee Comment (RC1) · Anonymous Referee #1 · 21 Oct 2019

**Review to "Ice nucleating ability of particulate emissions from solid biomass-fired cookstoves: an experimental study" by Korhonen et al. ACPD, 2019**

The manuscript by Korhonen et al. presents a laboratory investigation of the ice nucleation ability of particles emitted by burning different biomass types (fuels) in different types of cookstoves. Both the cookstoves and the fuel material represent typical material used in sub-Sahran Africa.

The ice nucleation ability of the emitted soot particles was tested in a commercial continuous flow diffusion chamber (SPIN). Ice nucleation experiments were performed in two different sets of experiments, that the author refer to as transient and chamber experiments.

In the transient experiments, RH-scans were performed at T = -32 °C and -28 °C in both deposition and immersion/condensation freezing experiments. In the chamber experiments, T-scans were performed at constant $RH_w$ = 115 %, measuring the ice nucleation activity in the immersion/condensation mode within a temperature range from approximately -32°C to -41 °C, using size-selected aerosol particles of different sizes. The setup further involves various instrument for particle characterization, such as an SP-AMS.

The main conclusion from the manuscript is that most of the soot types form ice at conditions slightly above those required for homogeneous freezing. Nevertheless, the statement in the conclusions that such particles emitted from biomass material in cookstoves can constitute a globally important source of INPs remains speculative in my opinion. A systematic comparison between the ice nucleation ability of the different soot particles sizes, fuel types and cookstoves types further remains difficult, given that not all fuel types and/or particle sizes were tested for all cookstoves. The authors also investigated different physico-chemical properties of the soot particles, but could not identify an evident correlation between those and the observed ice nucleation ability.

Overall, I find that the topic of the manuscript fits in the scope of ACP, given that the ice nucleation ability of anthropogenic aerosols such as soot remains an ongoing debate. Nevertheless, the manuscript in its current suffers from some inconsistencies (mostly technical), unnecessary repetitions and some issues that could be discussed in a clearer manner (see below). Overall, I suggest reevaluating the manuscript for publication in ACP after the points listed below have been critically addressed and incorporated.

**General comments:**

- In particular, the results sections contains some information that are required in the methods part. This hinders saliency of the most important findings when reading your Sect. 3. I encourage the authors to critically evaluate and improve the structure of the manuscript during revisions. For instance, a clearer separation of your transient and chamber experiments within Sect. 2.3 could be achieved by using subheadings. I would also encourage the authors to include a brief description of the auxiliary measurements made to characterize the particle physico-chemical properties in Sect. 2 and move the corresponding description of these there, which is currently in Sect. 3.2.
- Your current Sect. 3.2 is rather lengthy. I would consider breaking it up and have a separate section where you discuss e.g. your Fig. 7 (physico-chemical properties). If possible, please also add the particle characteristics for all your chamber experiments listed in Table 2. At the same time it remains unclear whether the "correlation" (L32, L412) is based on a visual inspection of the particle properties and the ice nucleation results, or whether it was quantified by e.g. calculating a correlation coefficient between the individual particle properties and e.g. the ice nucleation onset conditions discussed in the manuscript.
- It remains largely unclear to me how you estimate your "upscaling factors" based on your AS experiments, described in Sect. 3.3. This entire section needs to be improved to warrant a proper discussion of the biases of your SPIN results.

**Specific comments:**

- L24: I suggest to specify the sizes that were selected already in the abstract.
- L25: Please be more quantitative and add the exact temperature range used for SPIN experiments. "freezing conditions" is too vague.
- L32: Add "…and the observed increased ice nucleating ability."
- L33: Change to: "…aerosols from ice via immersion/condensation freezing at temperatures only moderately above homogeneous freezing conditions."

- L36: This statement should be followed by references. Consider adding: Korolev et al. (2017), Mülmenstädt et al. (2015).
- L36: "…Cloud droplets freeze…". This statement should be followed by a reference. You might want to give reference to standard textbooks, such as: Pruppacher and Klett (1997), Lohmann et al. (2016), Lamb and Verlinde (2011).
- L37: Please check the symbol of your degree Celsius units (°C), it seems off here and at other places in the text, e.g. L45, L64…
- L38: Please specific what MPC properties you mean. Do you mean radiative properties? See e.g. Matus and L'Ecuyer (2017).
- L41: Add: "number concentrations at the…"
- L43: You might want to add some never references giving overviews of INPs, e.g.: Kanji et al. (2017). At the same time, I think you should be more specific and give references of those studies that find soot to be a source of INPs, because citing Hoose and Möhler (2012) in this context seems to be in conflict with your citation on L46, please clarify.
- L46: Why do you only refer to immersion freezing? I think it would be better to say "immersion/condensation freezing" and more consistent with your usage further down, e.g. L64, and also with your statement on L224.
- L47: Change to: "…parameterisations used to estimate the soot-INP number concentration…"
- L49: It is not clear to me what you mean. Is the uncertainty 1 Wm$^{-2}$? Is this just the uncertainty of the indirect effects of soot particles on MPC, or are indirect effects on other cloud types (e.g. cirrus) also considered? Is this the *effective* radiative forcing (ERF), i.e. are fast adjustments included? It would be better practice to state the "best estimate" or "mean" ERF$_{aci}$ along with the uncertainty bounds and then make the statement that this is uncertain.
- L51: You also might want to cite: Bond et al. (2013)
- L53: Specify "… from incomplete combustion of fossil fuel or biomass burning. Properties of ambient soot particles…". Also add: Koehler et al. (2009), Mahrt et al. (2018)
- L53: Please further add references for how soot particle properties can change upon atmospheric aging. A study of interest could be: Bhandari et al. (2019).
- L54: Add: Ferry et al. (2002)
- L56: Please specify what you mean with "nanostructure".
- L60: Please quantify "significant source of particulate matter"
- L62: There is also evidence that particles from biomass burning are a source of INPs in cirrus clouds (e.g. DeMott et al. (2009)), which might be worth mentioning, given that there is increasing evidence, also from your paper, that any sort of combustion particles are an important source of INPs mainly (or only) in the cirrus regime at T < -38 °C.
- For the discussion of soot immersion freezing, you might want to add the following references to your discussion: Popovicheva et al. (2008), Brooks et al. (2014)
- L65: Change to: "has been observed to be associated…"
- L66: Change to: "…produced particles that acted as INPs at T = -35 °C, but not at T = -30 °C…"
- L68: Delete "they" and specify what you mean with "potentially significant climate impact".
- L69: This goes back to my comment on L60: Can you quantify the amount of PM emitted from biomass used for cooking vs. that emitted from natural forest fires, or at least give estimates of the order of magnitude?

- L71: What do you mean with "climate-forcing"? Do you mean a positive radiative forcing (warming)?
- L80: Add comma after cookstoves
- L89: Do not use capitalized letters for "water boiling test", to be consistent with L82 and change to "…(WBT, version 4.2.4), …"
- L91: Change "minutes" to min for consistency
- L92: It remains unclear whether the 45 min is the duration of the "simmering phase"or the time when the simmering phase starts, please clarify.
- L93: Please specify how this "height" affects your sampling. Does this height correspond to the orange squares in your Fig.1?
- L95: Change to "..diluted by 1:100.."
- L96: Change to "…of the analyzers."
- L104: Replace + by "and"
- L105: Please add the neutralizer type and the aerosol to sheath flow ratio you have operated your SMPS at.
- L118: Change "chapter" to Sect.
- L122: Change "experimentation" to experiments
- L124: Change to "…was approximately 2h before…"
- L124: How do you keep the particles suspended in the 15 $m^3$ chamber? Is there a fan at the bottom?
- L129: The mobility diameters of size selected particles for the IN experiments differs to those used for the CCN experiments (L109). Is there a reason for this?
- L134: Delete "simulated"
- L137: Change to: "Africa, which are commonly used…, and that represent regionally…"
- L138: Add: "For instance, the Casuarina…". Also, this statement should be followed by a reference.
- L148: Add: "located in the sub-…"
- L149: What do these "agricultural residues" encompass? Tree branches and leafs? Please clarify.
- L150: Consider rephrasing to: "…and investigations of using these residues to support the sustainable development of sub-Saharan are ongoing."
- L162: Change to: ", when forming cereal products"
- L165: Are you trying to say that the comparability results from comparing the African biomass fuels to something that has previously been studied and characterized, such as SW? Please clarify.
- L171: Change to "… (3S), which is a three…"
- L180: Replace version by gasifier stove
- L189: "more than five" is pretty arbitrary, I suggest to give a typical range here.
- L192/93: INP is already defined. I suggest to reformulate to: "Ice nucleation experiments where conducted with the spectrometer for ice nuclei (SPIN, DMT Inc., Colorado, USA), described by Garimella et al (2016)." There is no need to say since when it is operational in my eyes.
- L195: Please elaborate to what extend the temperature control was improved and how this affects your RH-uncertainty during your RH-ramps.
- L196: CFDC is already defined on L83.
- L196: Delete "with an IN-chamber design"
- L198/99: The abbreviations PINC and ZINC should be given in brackets.
- L200: Replace "ca." by approximately here and at other instances in text please.
- L203: Replace "before" with upstream
- L203: Add: "Therefore, the temperature of the…"

- L205: Write out: "…temperature, relative humidities and the path…"
- L210: Please specify where the background number comes from. I assume you refer to the OPC counts, right? Which optical particle size does this correspond to and how does this compare with the optical particle size used for calculating your activated fraction?
- L216: Delete "therefore". Also, equations should be used along normal/main text punctuation.
- L220: Add: "by the OPC, taking into account the dilution resulting from aerosol to sheath flow within SPIN…"
- L221: Usually the background measurement in CFDC measurements using RH-scans take place before the RH-scan, i.e. at low RH and after the RH-scan, i.e. at high RH. Is this how you corrected your sample? Please specify.
- L222: I would delete this statement here and simply add the Vali et al. (2010) reference to L223: "… is referred to as immersion freezing (Vali et al. 2010), …"
- L224: Change to: "… is expected prior to freezing."
- L229: Diluted from what initial number concentration?
- L230: This type of experiments is commonly referred to as "RH-scans" in the CFDC community.
- L231: Why do you say "scanning" here? Rephrase the sentence to: "were sampled at constant T= -32°C, scanning the $RH_i$…". Furthermore, it remains unclear why you chose -32°C for your RH-scans. This should be added in the text, along with a statement on the atmospheric relevance.
- L233: Why do you say "approximately" here but not before? If you have a fixed T and you know your RHi, then you can give "exact" RHw boundaries.
- L234: Delete: "and the only difference was in higher lamina temperature." This is clear the moment you state your T is -28 °C.
- L234/235: Delete statement or give reference to other studies.
- L237: Add : "combustion aerosol"
- L237: Specify "highly diluted" and give number concentration instead.
- L241: "T-ramp"
- L243 Why is the timer period for the first and the second background different (5 vs. 3 min)?
- L244: Do you mean -37.9 °C?
- L244: Why do you not show the activated fraction in the color bar of your Fig. 1b? This would be easier, as you use the activated fraction in the text. Also, having the x-axis of your Fig. 1b as lamina temperature would be more intuitive and would save you Fig. 1a, which in my eyes does not add much to the main findings of your study.
- L251: Replace "upward" by ascending for consistency
- L252: Change to: "Sect. 3.4"
- L236-252: It does not become entirely clear in this paragraph, but when looking at your Figs. 4, 5, 6 it looks to me that you fixed the lamina $RH_w$ of SPIN and then scanned different T, right? This can be referred to as "T-scan". This must become much clearer for the reader.
- L258: See my comment above: Showing your activated fraction or OPC counts as a function of lamina T, rather than time would be more appropriate.
- L245: How does this threshold of 4.5 microns compare to theoretical condensational growth calculations? Also, I do not understand why you need to discriminate these particles types, when using an evaporation section, as you describe on L203, please clarify. Do you refer to water droplet survival beyond the evaporation section?

- L259: Why are you using 6 microns as size threshold for ice crystals, if you argue above that the cloud droplets do not grow larger than 4.5 microns? Do you not introduce another bias then to your reported activated fractions? Please clarify in the text.
- L265: "since the ice crystal mode…" You might want to refer back to your Fig. 1 here.
- L269: Please specify "in a later section"
- L275: Delete "in real life"
- L276: Please clarify whether your ice counts are corrected by this factor. Which?
- L280-283: "Ice onset is defined.." This statement should be moved to the description of the methods. Moreover, I find your terminology of "negative" and "positive" results quite misleading. For your T-scan experiments (fixed RH): In case you observe ice formation at T > -38°C (or your experimentally defined homogeneous freezing threshold), refer to it as heterogeneous ice formation. Similarly, in your RH-scan experiments that take place at T = -32 °C, you are above the homogeneous nucleation temperature, so any ice formation you observe should be heterogeneous.
- L286: Please define "ultrafine", also for usage further down, e.g. L329.
- L289-L291: You need to elaborate on various statements made here:
    o The number concentration introduced into SPIN seems very high. Considering your aerosol to sheath flow ratio, you still have around 250 #/cc entering the OPC downstream of SPIN. What is the upper number concentration that can be sampled without co-incidence error?
    o It remains unclear how and why the detection limit depends on the particle number concentration introduced into the chamber, as you write in the caption of your Tab. 1. Could it be that at higher number concentrations introduced into SPIN you have relatively more particles > 250 nm (> 5%, as stated on L287) that become detected in your OPC? Please see also my comment to L210.
    o Your detection limit of the order $\alpha = 10^{-6}$ seems very low. It would be helpful to add an example plot for one of these experiments, showing $\alpha$ as a function of lamina RH in SPIN to the appendix. That way, the reader could more easily see at what RH you just sample noise and/or aerosol particles and at what point you actually start to see an ice signal. If $\alpha = 10^{-6}$ is your true detection limit, I suggest to more clearly write this in your statement on L211.
- L291: "Ice crystals…". This statement is not true looking at your "FD gasifier SW" experiment, if your threshold is $\alpha = 10^{-6}$.
- L291-294: Improve statement by saying: ""… four stoves do not nucleate ice in neither deposition nor immersion/condensation mode…"
- L297-318: This description should all be moved into your Sect. 2, where you describe your experimental methodology. Also, avoid unnecessary repetitions such as the mass concentration in the chamber (L304 vs. L122) to improve readability.
- L323: Your statement on the soot sizes selected and your typical number concentrations differs from what is given on L129, please clarify. Furthermore, the number concentration of 50-200 #/cc significantly differs to the number concentration used for the transient experiments (see L290), please comment.
- L323: Please quantify the "minor contributions of multiply charged particles", so that the reader can judge on how your reported activated fractions are affected. See Wiedensohler (1988). I suggest adding sizes and fractions of double charged particles to your Table 2.

- L324: "...which indicates that soot…" So do you only consider particles > 250 nm as soot, please clarify. If so, what are all the other particles? Please see also my comment on L323, where you say that you investigated 200 nm particles.
- L329: Replace "when" by "after"
- L331: Replace "Chapter" by "Sect." Please read the guidelines of ACP here: https://www.atmospheric-chemistry-and-physics.net/for_authors/manuscript_preparation.html
- On the same note of abbreviations, please also use "Fig." instead of "Figure" throughout the text.
- L333: Change to "ice activation", to be consistent with e.g. L336. In any case, please check for consistent spelling throughout manuscript. Moreover, I feel this paragraph seems to refer to your Table 3, right? In this case, having a reference to this table at the beginning of this discussion might be helpful.
- L333: Please be more careful in specifying which of the 3S experiments you talk about. For instance, the 3S on 500 nm CAS (see Table 3), shows that the ice onset is detected at T = -37.8 °C, rather than at -38.1 °C, as you write on L333.
- L337: "It is not clear at which temperature…" This statement seems misleading. Is this not why you do your AS calibration results, to determine homogenous freezing conditions?
- L339-L341: Why do you talk about the biases of the activated fraction here, when above (L330) you say that this is discussed in Sect. 3.4. I suggest to move this to the discussion around L436. Please improve the structure.
- L342: Please elaborate how you estimate this correction activated fraction of $10^{-3}$ for the temperature interval indicated.
- L349: Replace "as was done by" by "following"
- L349: "The uncertainty…" This sentence is a direct repetition of the previous statement, please delete.
- L353: Change "experiment emissions" to "experiments" and "high experiments" by "Good"
- L354: Change to "Sect. 3.4"
- L355: This statement needs to be clarified:
    o I cannot infer any size dependence from your Fig. 4. In other words, all the experimental curves for the 3S experiments are within T-uncertainty of each other. I encourage you to also add α-uncertainty bars, maybe for every third data point for clarity/visibility.
    o Can you verify a difference in chemical composition between the runs using the SP-AMS, you indicate in your Fig. 1? If there is a chemical difference between the 3S CAS 300 nm and 500 nm experiment, it is meaningful to compare the ice nucleation activity of different sizes of these aerosol types?
- L358: Add "than the ones…"
- L359: I cannot find the number "-37.7°C" in the RS experiments in your Tab. 3, please check.
- L361: "…and conditions for $10^{-3}$ ice-activated fraction were reached at -37.8 °C…" I think you do not need to repeat your ice nucleation onset activated fraction in every other statement. Just referring to "ice nucleation onset" would be fine (after you clearly defined it) and would improve the reading flow of your manuscript by a lot.
- L365: I would tune this down a little bit and say something along the lines of: "show that the soot particles emitted from burning the various fuels can act as heterogeneous INP at temperatures only slightly higher than those needed for homogeneous freezing of solution droplets."

- L368-377: "This indicates that very minor changes in combustion conditions significantly influence the ice nucleating ability of freshly produced soot particles." I very much agree and I think this is a great finding, however, I think this paragraphs deserves some more clarifications:
    o You say that you block one air supply, so your combustion should become less efficient, i.e. the organic carbon fraction of your soot particles should increase. Do you observe this in your SP-AMS data?
    o On L371 you say that you varied the pot height (see also L92). However, it remains unclear, how this was changed, whether this was a systematic change and how this affects the combustion process:
        ▪ For instance, I find it interesting that you seem to see a very strong change in ice nucleation ability between "FDGS, mod. 400 nm #1" and "FDGS, mod. 400 nm #2" in your Fig. 6 (which you interpret as change in chemical composition due to different combustion conditions), but hardly no change between "FDGS, mod. 400 nm #1" and "FDGS, mod. 500n m". Can you explain this?
        ▪ On the same note there is a strong size dependence for the 450 nm and 500 nm FDGS samples with blocked secondary air supply, but no size dependence between the "FDGS, mod. 400 nm #2" and the "FDGS, mod. 500nm". Can you elaborate on this?
        ▪ Looking at your Fig. 7 it is also interesting to note that the FDGS samples with blocked are supply for the 450 and 500 nm case seem to be quite different in their properties, e.g. Org/BC. How reproducible are your soot test points? In the end, what looks like a size dependence in Fig. 6, might also be caused by differences in physico-chemical properties.
- L382: Why is the effective density of 350 nm particles representative for your aggregates of different sizes? The effective density is a strong function of particle size, see e.g. Olfert et al. (2017), as you also note on L392.
- L383: There is no Fig. 8.
- L383-L391: This should be mentioned in your methods section along with a more detailed description and/or reference, how you calculate the chemical properties from the SP-AMS data, not in the results section.
- L391: "The values of…" Is this true for any soot aggregate size? Please specify.
- L396: "OA" is not defined. You might also want to use "OA" instead of "Org" in your Fig. 7.
- L396: Should not K400#1 have a higher UV absorption, given the high Org carbon fraction?
- L412: Can you quantify this correlation?
- L416: "Further studies…" I suggest to move this statement to your Sect. 4.
- L418: "The typical…" This is a repetition from L250 and can be deleted.
- L425: Change "Chapter" to "Sect."
- L426: Replace "to" by "the"
- L428: This is a direct repetition of L346 and can be deleted.
- L430-434: I think that it would be best practice to report the error corresponding to the largest uncertainty, i.e. the span in T across the lamina ("T profile") rather than the variation in mean lamina T. See e.g. Garimella et al. (2017)
- L441: What does "somewhat similar operation conditions" mean? You should clarify this.
- L445-461: This paragraph need to be significantly elaborated and improved in order to justify a proper discussion of biases in ice crystal detection.

- o You list a bunch of reasons for underestimating you activated fraction (e.g. choice of 6 μm channel for ice detection), but you lack to quantify the contributions of this to the activated fractions reported here.
  - o It remains unclear why the "underestimation depends on lamina temperature".
  - o How do you arrive at the upscaling factors of 3-5 and 5-10? Please add this to the text.
  - o Finally, please be more quantitative when saying "in a neighborhood of -40 °C"
- L468: "This study shows…" It remains unclear whether this statement refer to all different cook stoves tested or not.
- L469: "500 million people…" This should be followed by a reference and I would move this statement to the introduction, as it does not constitute a conclusion from your study.
- L471: "…showed good performance". This statement is misleading, given that your maximum activated fractions even for the AS experiments are 90% below the theoretically expected values, please be more specific.
- L475: It would be good to quantify "poor IN activity" (not INP activity!) by giving a range of observed activated fractions.
- L481: "Therefore…" Is this really true? Do all the cook stove emission be transported upwards? What about the atmospheric lifetime of these aerosols and what are the cloud properties that you say become changed?
- L487-489: This comparison is very loose and insubstantial and requires a more adequate discussion and comparison of dust/soot emissions, (vertical) burdens, atmospheric lifetimes and many more factors. In the end, your results show that most of the soots can nucleate ice only very close to conditions required for homogeneous freezing. As such, the impact of these soot types on warm or MPCs is very likely absent and/or negligible. In fact, when one takes -38 °C as a "general threshold" for homogeneous freezing most of your soots in Figs. 4 and 5 freeze homogeneously
- L490: Delete "of"
- L499: What do you mean by "slight differences"? Which of the "studied properties" were different? Please be more precise and quantitative here, otherwise it is hard for the reader to take out the main findings of your study.
- L761: Change "," to "." In front of "The two…"
- L767: The dependence of the detection limit on the sample concentration should be discussed in the main text. If this is the case, it would also be meaningful to add the concentrations for the individual experiments listed here.
- L779: Add "are equal…"

Bhandari, Janarjan, et al. (2019), 'Extensive Soot Compaction by Cloud Processing from Laboratory and Field Observations', *Scientific Reports,* 9 (1), 11824.

Bond, T. C., et al. (2013), 'Bounding the role of black carbon in the climate system: A scientific assessment', *Journal of Geophysical Research-Atmospheres,* 118 (11), 5380-552.

Brooks, S. D., Suter, K., and Olivarez, L. (2014), 'Effects of chemical aging on the ice nucleation activity of soot and polycyclic aromatic hydrocarbon aerosols', *J Phys Chem A,* 118 (43), 10036-47.

DeMott, P. J., et al. (2009), 'Ice nucleation behavior of biomass combustion particles at cirrus temperatures', *Journal of Geophysical Research-Atmospheres,* 114 (D16), D16205.

Ferry, D., et al. (2002), 'Water adsorption and dynamics on kerosene soot under atmospheric conditions', *Journal of Geophysical Research-Atmospheres,* 107 (D23).

Garimella, S., et al. (2017), 'Uncertainty in counting ice nucleating particles with continuous diffusion flow chambers', *Atmos. Chem. Phys. Discuss.,* 2017, 1-28.

Hoose, C. and Möhler, O. (2012), 'Heterogeneous ice nucleation on atmospheric aerosols: a review of results from laboratory experiments', *Atmospheric Chemistry and Physics,* 12 (20), 9817-54.

Kanji, Zamin A., et al. (2017), 'Overview of Ice Nucleating Particles', *Meteorological Monographs,* 58 (0), 1.1-1.33.

Koehler, Kirsten A., et al. (2009), 'Cloud condensation nuclei and ice nucleation activity of hydrophobic and hydrophilic soot particles', *Physical Chemistry Chemical Physics,* 11 (36), 7906-20.

Korolev, A., et al. (2017), 'Mixed-Phase Clouds: Progress and Challenges', *Meteorological Monographs,* 58, 5.1-5.50.

Lamb, Dennis and Verlinde, Johannes (2011), *Physics and Chemistry of Clouds* (Cambridge University Press).

Lohmann, Ulrike, Lüönd, Felix, and Mahrt, Fabian (2016), *An Introduction to Clouds: From the Microscale to Climate* (1st edition edn.; Cambridge: Cambridge University Press).

Mahrt, F., et al. (2018), 'Ice nucleation abilities of soot particles determined with the Horizontal Ice Nucleation Chamber', *Atmospheric Chemistry and Physics,* 18 (18), 13363-92.

Matus, A. V. and L'Ecuyer, T. S. (2017), 'The role of cloud phase in Earth's radiation budget', *Journal of Geophysical Research-Atmospheres,* 122 (5), 2559-78.

Mülmenstädt, Johannes, et al. (2015), 'Frequency of occurrence of rain from liquid-, mixed-, and ice-phase clouds derived from A-Train satellite retrievals', *Geophysical Research Letters,* 42 (15), 6502-09.

Olfert, Jason S., et al. (2017), 'Effective density and volatility of particles sampled from a helicopter gas turbine engine', *Aerosol Science and Technology,* 51 (6), 1-11.

Popovicheva, O., et al. (2008), 'Effect of soot on immersion freezing of water and possible atmospheric implications', *Atmospheric Research,* 90 (2), 326-37.

Pruppacher, H. R. and Klett, D. J. (1997), *Microphysics of Clouds and Precipitation* (2nd edition edn.; Dordrecht, The Netherlands: Kluwer Academic Publishers).

Wiedensohler, A. (1988), 'An approximation of the bipolar charge-distribution for particles in the sub-micron size range', *Journal of Aerosol Science,* 19 (3), 387-89.

-

---

## Referee Comment (RC2) · Anonymous Referee #2 · 8 Dec 2019

Ice nucleation activity of soot particles from several biomass fuels was investigated. The physical and chemical properties of these soot particles were also studied. These measurements are very useful to understand the implications of soot emissions from solid fuels that are very commonly used worldwide. Such INP data is scarce, and I recommend 'publication' after addressing the following comments.

1. To better understand the implications of these measurements, it would be best convert the data shown in Figures 4 to 6 to active site density (ns) or active fraction kind of metric and compare against other INP data (soot, dust, etc.) from literature. This will help to put the data in the context of other INPs.

[Figure]

2. Figures and Tables. In Figure 1, do the ejector dilution (ED) also helps to cool the samples? (this is described on line 95). It is not clear how ice crystals can grow to size up to 11 um (Figure 2) as droplets only grew to 4.5 um only (page 256). Would you please explain this observation? If these droplets freeze, the size of the ice crystal should be equal to the droplet size, correct? From ~20.5 to 20.6 hrs (Figure 2), where ice crystals are observed, there are some particles of size 2 ± 1 um observed. How is this possible? All the droplets should be frozen at this temperature. If these are not droplets, then why such small ice crystals are observed? Please elaborate caption of figure 3. SPIN data is not shown here. How is the contribution from multiple charge particles is corrected for the data shown in figure 3? SPIN was operated at RHw = 115% and without depol. detector. How was droplet breakthrough artifact addressed? It is not clear if the ice activation threshold is 1e-3, then how data is shown up to 1e-4 (see figures 4 to 6). If the data (from 1e-3 to 1e-4) is not trustworthy because of the threshold limit, I would revise the figures to show data from 1e-3 to 1 only. Please explain what X-axis labels in Figure 7 are. What is K500? How these labels are related to figures 4 to 6. I think there is a typo ('ja') on line 762. The ice detection limit (figure 4 to 6) shows 1e-3, but in Table 1 detection limit is in the range of 1e-06. Please clarify this discrepancy and definition of the detection limit. From a readability perspective, it would be better to spell out the abbreviations (e.g., ND, FD etc.) that appeared in Tables 1 to 3 in the Table caption-text itself.

3. It is mentioned that the evaporation section is not efficient (line 262). Would you please explain this feature of the SPIN. Why it is not efficient, how it affects the data presented here, and how it is operated differently from other SPIN and CFDC style chambers. A paragraph from Line 273 to 276. Please elaborate on this argument. Does the correction factor was applied? If yes, how this factor was determined. There is some discussion in section 3.3; however, it is not very clear. The factor estimates described on line 458 to 461 are not proved using INP measurements. These are speculations. Please justify.

---

## Author Comment (AC1) · 7 Feb 2020

**Author response to comments by Referee #1**

We thank Referee #1 for his/her work on the manuscript, and comprehensive comments and suggestions on revisions how to improve the contents. The detailed responses (written on red font) to comments as they were posted (green font) are listed below – unless the response includes how the text has been revised by itself, the revisions to the text are marked separately on blue font.

**General comments:**

In particular, the results sections contains some information that are required in the methods part. This hinders saliency of the most important findings when reading your Sect. 3. I encourage the authors to critically evaluate and improve the structure of the manuscript during revisions. For instance, a clearer separation of your transient and chamber experiments within Sect. 2.3 could be achieved by using subheadings. I would also encourage the authors to include a brief description of the auxiliary measurements made to characterize the particle physico-chemical properties in Sect. 2 and move the corresponding description of these there, which is currently in Sect. 3.2.

We acknowledge the mentioned structural shortcomings and have moved all methodology information to Sect 2. We have also shortened lengthy sections and added subsections, separating the descriptions of the CFDC, ice nucleation experiments (transient and chamber) and particle characterization, each to their own respective section.

Your current Sect. 3.2 is rather lengthy. I would consider breaking it up and have a separate section where you discuss e.g. your Fig. 7 (physico-chemical properties). If possible, please also add the particle characteristics for all your chamber experiments listed in Table 2. At the same time it remains unclear whether the "correlation" (L32, L412) is based on a visual inspection of the particle properties and the ice nucleation results, or whether it was quantified by e.g. calculating a correlation coefficient between the individual particle properties and e.g. the ice nucleation onset conditions discussed in the manuscript.

We agree and have revised the text, adding a separate section for emission particle properties.

There is a tendency of the two more ice-active samples having a relatively lower ratio. However, the differences between the samples are not pronounced considering the errors, and we do not observe a significant correlation with this limited data set.

It remains largely unclear to me how you estimate your "upscaling factors" based on your AS experiments, described in Sect. 3.3. This entire section needs to be improved to warrant a proper discussion of the biases of your SPIN results.

Please see our responses to the specific comments relevant to discussion on the upscaling factor.

**Specific comments:**

- L24: I suggest to specify the sizes that were selected already in the abstract.

A statement that specifies the sizes has been added:

"The soot particles in nm were 250, 260, 300, 350, 400, 450 and 500."

- L25: Please be more quantitative and add the exact temperature range used for SPIN experiments. "freezing conditions" is too vague.

The text has been changed from:

"to water-supersaturated freezing conditions in the SPIN."

to:

"introduced to water-supersaturated freezing conditions (-32 °C to -43 °C) in the SPIN."

- L32: Add "…and the observed increased ice nucleating ability."

Statement added.

- L33: Change to: "…aerosols from ice via immersion/condensation freezing at temperatures only moderately above homogeneous freezing conditions."

Done.

- L36: This statement should be followed by references. Consider adding: Korolev et al. (2017), Mülmenstädt et al. (2015).

References added.

- L36: "…Cloud droplets freeze…". This statement should be followed by a reference. You might want to give reference to standard textbooks, such as: Pruppacher and Klett (1997), Lohmann et al. (2016), Lamb and Verlinde (2011).

A reference to Pruppacher and Klett (1997) added.

- L37: Please check the symbol of your degree Celsius units (°C), it seems off here and at other places in the text, e.g. L45, L64…

Symbols checked and corrected throughout the manuscript.

- L38: Please specific what MPC properties you mean. Do you mean radiative properties? See e.g. Matus and L'Ecuyer (2017).

A wide range of cloud properties are influenced by introducing an ice phase in a liquid phase supercooled cloud. For example, the formation of ice releases latent heat, the presence of an ice phase influences the vapour pressures, and mixed-phase clouds are not in thermodynamic equilibrium. The growth of ice crystals on the expense of liquid droplets influence the cloud optical properties. To clarify this, we have modified the text (L38) from:

"MPC properties and lifetime are very sensitive to the formation of solid ice,"

To:

"A wide range of MPC properties including radiative properties and lifetime are sensitive to the formation of solid ice (Matus and L'Ecuyer, 2017),"

- L41: Add: "number concentrations at the…"

The text has been changed from:

"but there are significant gaps in our knowledge regarding these important ice formation processes."

to:

"but there are significant gaps in our knowledge regarding INP concentrations within the MPCs and these important ice formation processes."

- L43: You might want to add some never references giving overviews of INPs, e.g.: Kanji et al. (2017). At the same time, I think you should be more specific and give references of those studies that find soot to be a source of INPs, because citing Hoose and Möhler (2012) in this context seems to be in conflict with your citation on L46, please clarify.

Suggested reference added.

It is outside the scope of our study to review all experimental ice nucleation studies involving soot particles because there are too many of them and it would only be of value providing information about the particles studied. Hoose and Möhler (2012) provided a nice overview of many studies including positive as well as negative ice nucleation results for soot particles, and the reader can find further information there. Hence, there is no conflict in using that citation in both contexts: soot particles showing no or a high heterogeneous ice nucleating ability in the immersion/condensation mode. We have chosen to cite three specific studies with positive results in L43-45, and quite many additional studies in other parts of this manuscript.

We have added the Levin et al. (2016) study to L65, and thus we cite all relevant immersion/condensation mode ice nucleation studies on biomass burning aerosol including soot particles, which we know of in L. 62-67.

We do find that the references included in the revised manuscript are balanced according to the aim and context of our study – without citing all ice nucleation studies on soot particles directly.

- L46: Why do you only refer to immersion freezing? I think it would be better to say "immersion/condensation freezing" and more consistent with your usage further down, e.g. L64, and also with your statement on L224.

Added "immersion and condensation freezing" according to suggestion.

- L47: Change to: "…parameterisations used to estimate the soot-INP number concentration…"

Done.

- L49: It is not clear to me what you mean. Is the uncertainty 1 Wm-2? Is this just the uncertainty of the indirect effects of soot particles on MPC, or are indirect effects on other cloud types (e.g. cirrus) also considered? Is this the effective radiative forcing (ERF), i.e. are fast adjustments included? It would be better practice to state the "best estimate" or "mean" ERFaci along with the uncertainty bounds and then make the statement that this is uncertain.

What we wish to emphasize in the context of our study, is that the current climate models are highly sensitive to the 'choice' of the ice nucleating ability of ambient soot particles. The range in RF (0.111 to 1.059Wm−2) reported by Yun et al. (2013) reflect the result of a low versus a high ice nucleation impact of ambient soot particles on MPCs. We have modified the text from:

"The radiative forcing associated with the impact of soot particles on MPCs has been reported to be very uncertain and potentially up to about 1 Wm-2 (Yun et al., 2013)."

To:

"The radiative forcing associated with the impact of fossil fuel soot particles on MPCs has been reported to range from about 0.1 to about 1 Wm$^{-2}$ depending on their ice nucleating ability, which is uncertain (Yun et al., 2013)."

- L51: You also might want to cite: Bond et al. (2013)

Reference added.

- L53: Specify "… from incomplete combustion of fossil fuel or biomass burning. Properties of ambient soot particles…". Also add: Koehler et al. (2009), Mahrt et al. (2018)

The text has been changed from:

Soot particles are produced from incomplete and ambient soot particle properties are highly variable and influenced by the combustion conditions and atmospheric ageing (Corbin et al., 2015).

to:

Soot particles are produced from incomplete combustion (combustion with insufficient oxygen supply) and ambient soot particle properties, such as morphology and chemical composition, are highly variable and influenced by the combustion conditions and atmospheric ageing (Ferry et al, 2002; Popovicheva et al., 2008; Koehler et al., 2009; Corbin et al., 2015; Mahrt et al., 2018; Bhandari et al., 2019).

- L53: Please further add references for how soot particle properties can change upon atmospheric aging. A study of interest could be: Bhandari et al. (2019).

Bhandari et al., 2019 added to references. Please see previous response regarding L53.

- L54: Add: Ferry et al. (2002)

Ferry et al., 2002 added to references.

- L56: Please specify what you mean with "nanostructure".

The nanostructure refers to characteristics on a 'nano-scale' such as fringe length and structural order. Häusler et al. (2018) show that the ordering of graphene on a nanoscale influences the ice nucleating ability. We have modified the text (L56-57) from:

"Chemical groups on the soot particle surface with the ability to form hydrogen bonds with water molecules are likely to be of importance (Gorbunov et al., 2001) as well as the soot particle nanostructure (Häusler et al., 2018)."

To:

"Chemical groups on the soot particle surface with the ability to form hydrogen bonds with water molecules are likely to be of importance (Gorbunov et al., 2001). In addition, the soot particle nanostructure may also be of importance, with highly ordered graphene structures being more efficient in supporting ice nucleation relative to lowly ordered graphene structures (Häusler et al., 2018)."

- L60: Please quantify "significant source of particulate matter"

The text has been changed from:

"Biomass combustion for cooking is an important global source of energy, being the major environmental health risk worldwide, as it is a significant source of particulate matter (PM) and soot particles on regional to global scales (Lim et al., 2012; Bonjour et al., 2013)."

to:

"Biomass combustion for cooking is an important global source of energy, being also the major environmental health risk worldwide (Lim et al., 2012). It has been estimated that approximately 2.8 billion people depend on solid fuel combustion for daily cooking worldwide, mostly in the developing part of the World (Bonjour et al., 2013). Therefore, solid biomass combustion is a significant source of particulate matter (PM) and soot particles on regional to global scales."

- L62: There is also evidence that particles from biomass burning are a source of INPs in cirrus clouds (e.g. DeMott et al. (2009)), which might be worth mentioning, given that there is increasing evidence, also from your paper, that any sort of combustion particles are an important source of INPs mainly (or only) in the cirrus regime at T < -38 °C.

DeMott et al. (2009) reported all freezing events to be indistinguishable from homogeneous freezing. Hence, we do not find direct evidence in that study of such a link. However, the findings presented in that study do not exclude potential heterogeneous freezing e.g. for the aged aerosol biomass burning particles.

- For the discussion of soot immersion freezing, you might want to add the following references to your discussion: Popovicheva et al. (2008), Brooks et al. (2014)

References added to respective relevant contexts.

- L65: Change to: "has been observed to be associated..."

Done.

- L66: Change to: "...produced particles that acted as INPs at T = -35 °C, but not at T = -30 °C..."

Done.

- L68: Delete "they" and specify what you mean with "potentially significant climate impact".

The text has been changed from:

"Huang et al. (2018) modeled the indirect climate impact of solid fuelled cookstove aerosol emissions, and they reported a potentially significant climate impact from emitted soot particles on MPC conditions and thus climate."

to:

"Huang et al. (2018) modeled the indirect climate impact of solid fueled cookstove aerosol emissions and reported a potentially significant climate impact in increase of high clouds in their model runs where black carbon (BC) particles acted as INPs. Therefore, soot particle emissions can potentially affect MPC conditions and thus climate."

- L69: This goes back to my comment on L60: Can you quantify the amount of PM emitted from biomass used for cooking vs. that emitted from natural forest fires, or at least give estimates of the order of magnitude?

Such estimates are presented by Huang et al. (2018), to which the statement refers to. According to their study, the black carbon (BC) emissions from biomass-fired cookstoves are globally 2.31 Tg yr$^{-1}$, which accounts for 23.7% of total BC emissions. Primary organic matter (POM) emissions from cookstoves account for 21% of total global POM emission budget. We consider this an appropriate quantification of global cookstove emissions.

- L71: What do you mean with "climate-forcing"? Do you mean a positive radiative forcing (warming)?

Yes, the statement is corrected from:

"However, an efficient utilization of modern biomass fuels in efficient biomass cookstoves can constitute an alternative to mitigate climate-forcing aerosol emissions and household air pollution (HAP) worldwide (Carvalho et al. 2016)."

to:

"However, an efficient utilization of modern biomass fuels in efficient biomass cookstoves can constitute an alternative to mitigate household air pollution (HAP) worldwide, and aerosol emission effect on radiative forcing (Carvalho et al. 2016)."

- L80: Add comma after cookstoves

Done.

- L89: Do not use capitalized letters for "water boiling test", to be consistent with L82 and change to "…(WBT, version 4.2.4), …"

Done.

- L91: Change "minutes" to min for consistency

Done.

- L92: It remains unclear whether the 45 min is the duration of the "simmering phase"or the time when the simmering phase starts, please clarify.

Yes, the statement is corrected from:

"A typical simulation time was 60-85 min in total, including two phases of the WBT process: cold start (15-40 min) and simmering (45 min)."

to:

"A typical simulation time was 60-85 min in total, including two phases of the WBT process: cold start (phase duration 15-40 min) and after that simmering (duration 45 min)."

- L93: Please specify how this "height" affects your sampling. Does this height correspond to the orange squares in your Fig.1?

Figure 1 shows the Three-stone fire as an example, and the orange squares do represent bricks on which the cooking pot was placed during the boiling tests. For other cookstoves, the cooking pot was normally placed on top of the cookstove: this type of operation represents usage of the cookstoves in the way they have been designed to, by manufacturers of commercially available cookstoves (Rocket stove, ND gasifier and FD gasifier). The three special experiments revealed that

changing the height from designated one can increase the particle emissions, as L311-315 stated later in the text.

- L95: Change to "..diluted by 1:100.."

Done.

- L96: Change to "…of the analyzers."

Done.

- L104: Replace + by "and"

Done.

- L105: Please add the neutralizer type and the aerosol to sheath flow ratio you have operated your SMPS at.

The sentence describing the SMPS system has been changed from:

"An SMPS system (classifier, TSI 3082 + condensation particle counter, TSI 3775) measured particle size distribution and concentration in chamber experiments at scan range from 15.7 nm to 615.3 nm, with continuously repeated 180-second scans during aerosol injection and sampling."

to:

"An SMPS system (classifier TSI 3082 equipped with aerosol neutralizer TSI 3088, and condensation particle counter, TSI 3775) measured particle size distribution and concentration in chamber experiments at scan range from 15.7 nm to 615.3 nm, with continuously repeated 180-second scans during aerosol injection and sampling at an aerosol to sheath flow ratio of 0.3 vs. 3 litres per min, respectively."

- L118: Change "chapter" to Sect.

Done.

- L122: Change "experimentation" to experiments

Done.

- L124: Change to "…was approximately 2h before…"

Done.

- L124: How do you keep the particles suspended in the 15 m3 chamber? Is there a fan at the bottom?

Yes, there was a fan at the bottom of the storage chamber to ensure mixing. Gravitational settling was not a significant issue for the stored aerosol populations within the time frames of these experiments of typically less than 3 hours.

- L129: The mobility diameters of size selected particles for the IN experiments differs to those used for the CCN experiments (L109). Is there a reason for this?

Yes. The CCNC used a fixed scheme for electrical mobilities 65 nm, 100 nm, 200 nm and at times 350 nm when available in sufficient number concentrations. The IN experiments always aimed for

sampling the largest particle size that was available in number concentrations larger than 30 cc$^{-1}$, which was stated on L322.

- L134: Delete "simulated"

Done.

- L137: Change to: "Africa, which are commonly used…, and that represent regionally…"

Done.

- L138: Add: "For instance, the Casuarina…". Also, this statement should be followed by a reference.

Reference added. The text has been changed from:

"The *Casuarina equisetifolia* (CAS), commonly known as Horsetail tree, is a species which is commonly used in agroforestry systems in the southern part of the African continent."

to:

"For instance, the *Casuarina equisetifolia* (CAS), commonly known as Horsetail tree, is a species which is commonly used in agroforestry systems in the southern part of the African continent (Potgieter et al., 2014)."

- L148: Add: "located in the sub-…"

Done.

- L149: What do these "agricultural residues" encompass? Tree branches and leafs? Please clarify.

The statement is corrected from:

"During processing, agricultural residues form up to 50% of the total weight of coffee products (Oliveira and Franca, 2015) and the usage of these residues as biofuels is under research for potential sustainable development in sub-Saharan Africa."

to:

"During processing, agricultural residues such as tree branches and leaves form up to 50% of the total weight of coffee products (Oliveira and Franca, 2015) and the usage of these residues as biofuels is under research for potential sustainable development in sub-Saharan Africa."

- L150: Consider rephrasing to: "…and investigations of using these residues to support the sustainable development of sub-Saharan are ongoing."

Done.

- L162: Change to: ", when forming cereal products"

Done.

- L165: Are you trying to say that the comparability results from comparing the African biomass fuels to something that has previously been studied and characterized, such as SW? Please clarify.

Yes, the SW pellets are a standardized product whose characteristics, such as composition and origin of raw material, are known.

- L171: Change to "… (3S), which is a three…"

Done.

- L180: Replace version by gasifier stove

Done.

- L189: "more than five" is pretty arbitrary, I suggest to give a typical range here.

The statement is corrected from:

"A typical number of re-fueling events was more than five."

to:

"A typical number of re-fueling events was between five and ten."

- L192/93: INP is already defined. I suggest to reformulate to: "Ice nucleation experiments where conducted with the spectrometer for ice nuclei (SPIN, DMT Inc., Colorado, USA), described by Garimella et al (2016)." There is no need to say since when it is operational in my eyes.

Statement reformulated according to suggestion and mentioning about the operational period has been removed.

- L195: Please elaborate to what extend the temperature control was improved and how this affects your RH-uncertainty during your RH-ramps.

The main improvement was that the ice nucleation chamber was re-designed to reduce temperature variation on different spots on its walls, and thus improve control of the lamina temperature and RH. Also additional thermocouples that monitor the temperature on different were installed, which enabled optimized temperature and *RH* control during the scans.

The statement is corrected from:

"The main difference relative to the older version used by Ignatius et al. (2016) is improved temperature control."

to:

"The main difference relative to the older version used by Ignatius et al. (2016) is improved temperature control via e.g. by an increase in the total number of thermocouples from 8 to 32 in the main chamber (growth section)."

- L196: CFDC is already defined on L83.

Corrected.

- L196: Delete "with an IN-chamber design"

Done.

- L198/99: The abbreviations PINC and ZINC should be given in brackets.

Corrected.

- L200: Replace "ca." by approximately here and at other instances in text please.

Corrected throughout the manuscript.

- L203: Replace "before" with upstream

Done.

- L203: Add: "Therefore, the temperature of the…"

Done.

- L205: Write out: "…temperature, relative humidities and the path…"

Done.

- L210: Please specify where the background number comes from. I assume you refer to the OPC counts, right? Which optical particle size does this correspond to and how does this compare with the optical particle size used for calculating your activated fraction?

The following specification has been added to the text:

"Here, the background signal refers to unwanted OPC counts in the ice crystal size range, which may be due to break up of frost on the ice-covered plates or alternatively due to tiny leaks in the instrument chamber."

Typical background signal counts due to cracking ice layer represent particles that are larger than 6 µm in optical size, therefore the chamber was re-iced when the background signal exceeded 10-15 particles per litre, as is stated on L211.

- L216: Delete "therefore". Also, equations should be used along normal/main text punctuation.

Done. ":" added.

- L220: Add: "by the OPC, taking into account the dilution resulting from aerosol to sheath flow within SPIN…"

Done.

- L221: Usually the background measurement in CFDC measurements using RH-scans take place before the RH-scan, i.e. at low RH and after the RH-scan, i.e. at high RH. Is this how you corrected your sample? Please specify.

Yes, the background check was carried out just before and just after each RH scan, respectively. Linear interpolation was used to estimate the background during scans.

- L222: I would delete this statement here and simply add the Vali et al. (2010) reference to L223: "… is referred to as immersion freezing (Vali et al. 2010), …"

Done.

- L224: Change to: "… is expected prior to freezing."

Done.

- L229: Diluted from what initial number concentration?

Typical number concentrations after ejector dilution by 1:100 were on the order of $10^5$-$10^6$ cm$^{-3}$ that would had been excessive for a CFDC instrument.

- L230: This type of experiments is commonly referred to as "RH-scans" in the CFDC community.

Mentioning about RH-scan added.

- L231: Why do you say "scanning" here? Rephrase the sentence to: "were sampled at constant T= -32°C, scanning the RHi...". Furthermore, it remains unclear why you chose -32°C for your RH-scans. This should be added in the text, along with a statement on the atmospheric relevance.

A sample temperature of -32°C is commonly applied in CFDC studies (e.g. Levin et al., 2016). Measurements at relatively low temperatures improve the signal to noise ratio, and when no INPs are detected for such conditions – we can rule out significant INP concentrations at higher temperatures relevant for MPCs.

The text has been changed from:

"The measurement sequence used in the transient experiments was as follows:"

to:

"*RH*-scans were used for the transient experiments at constant *T* = -32°C, scanning the *RHi* from 100 % up to 160 %, which is close to the procedure used by Petters et al. (2009) and Levin et al., (2016)."

- L233: Why do you say "approximately" here but not before? If you have a fixed T and you know your RHi, then you can give "exact" RHw boundaries.

Word "approximately" deleted.

- L234: Delete: "and the only difference was in higher lamina temperature." This is clear the moment you state your T is -28 °C.

Done.

- L234/235: Delete statement or give reference to other studies.

Added references to Petters et al. (2009) and Welti et al. (2009), who both applied the RH-scan in their CFDC experiments.

- L237: Add : "combustion aerosol"

Done.

- L237: Specify "highly diluted" and give number concentration instead.

The term 'highly diluted' refers to the ammonium sulfate concentration in the formed cloud droplets (diameters in super-micron range with a seed diameter of 350 nm) in the upper part of the chamber. We suppose that the reviewer suggests including the sample particle number concentration.

The text has been changed from:

"This procedure is illustrated in Fig. 2, where a homogeneous freezing experiment on highly diluted ammonium sulfate (AS, dry mobility diameter of 350 nm) droplets is used as an example."

to:

"This procedure is illustrated in Fig. 2, where a homogeneous freezing experiment on highly diluted ammonium sulfate (AS, dry mobility diameter of 350 nm) droplets were introduced to the SPIN in number concentration of approximately 150 cm$^{-3}$."

- L241: "T-ramp"

Added.

- L243 Why is the timer period for the first and the second background different (5 vs. 3 min)?

The reason for this was that before the T-ramp, the CFDC was running on initial settings and sampled filtered air prior to start for at least 5 minutes before the T-ramp was initiated. The statement has been changed from:

"First, the internal background was checked via sampling filtered dry air for 5 minutes before the T-ramp began."

to:

"First, the internal background was checked via sampling filtered dry air for at least 5 minutes before the T-ramp began."

- L244: Do you mean -37.9 °C?

Yes. We observed that there is a difference in ice onset *T* between ascending and descending *T*-ramps, and possible reasons for this are discussed in paragraph.

- L244: Why do you not show the activated fraction in the color bar of your Fig. 1b? This would be easier, as you use the activated fraction in the text. Also, having the x-axis of your Fig. 1b as lamina temperature would be more intuitive and would save you Fig. 1a, which in my eyes does not add much to the main findings of your study.

We believe the reviewer refers to Fig. 2. It is not possible to include the activated fraction in Fig. 2b in a meaningful manner. In the figure, we show particle counts in bins without any ice crystals, as well as size bins dominated by ice crystals. Normalisation to the total particle concentration would neither provide an ice-activated fraction – nor more information to the figure.

We agree that Fig 2a alone does not add anything to the main findings of the study, but it shows very clearly why the upward temperature ramps provide significantly better control of the lamina conditions. That is of importance to present, since we are not familiar with similar operation of CFDCs with increasing T-ramps from previous studies.

It would indeed be possible to present fractions of Fig. 2b as a function of average lamina temperature instead of time – but then it would not be possible to include the background checks in a meaningful way, as the lamina temperature is close to constant during those time windows.

Since we operated our CFDC in an untraditional way, we do find it highly relevant to openly show this in Fig. 2, and we find that the relevant information is available from the figure in the current version.

- L251: Replace "upward" by ascending for consistency

Done.

- L252: Change to: "Sect. 3.4"

Done.

- L236-252: It does not become entirely clear in this paragraph, but when looking at your Figs. 4, 5, 6 it looks to me that you fixed the lamina RHw of SPIN and then scanned different T, right? This can be referred to as "T-scan". This must become much clearer for the reader.

Yes, that was the experiment procedure. References to T-scan added to the paragraph, for instance the ramp description has been edited from:

"All experiments were carried out on similar automated ramps containing the following steps: First, the internal background was checked via sampling filtered dry air for 5 minutes before the lamina T-ramp began."

to:

"All experiments were carried out on similar automated *T*-scan ramps containing the following steps: First, the internal background was checked via sampling filtered dry air for at least 5 minutes before the *T*-scan began."

- L258: See my comment above: Showing your activated fraction or OPC counts as a function of lamina T, rather than time would be more appropriate.

Please see our response to comment for L244.

- L245: How does this threshold of 4.5 microns compare to theoretical condensational growth calculations? Also, I do not understand why you need to discriminate these particles types, when using an evaporation section, as you describe on L203, please clarify. Do you refer to water droplet survival beyond the evaporation section?

We assume that the reviewer initially refers to the statement in L255 (and not L245). A subset of the SPIN experiments were intentionally carried out with water droplet survival as indicated e.g. in Fig. 2. With that operation mode, the mode of droplets surviving the evaporation section never grew larger than ~4.5 micrometers, which allowed us to consider any optically larger particles as ice crystals. The motivation for this mode of operation was to ensure that the vast majority of size selected soot particles in the lamina were exposed to a supersaturation with respect to water sufficient for cloud droplet activation, which allowed for investigation of condensation/immersion mode freezing. As shown by Garimella et al.(2016), the evaporation section of SPIN does not allow for operation at high supersaturations with respect to water before droplet breakthrough is observed. This 'new' operation approach (allowing droplet breakthrough) leads to the ice-activated fraction being biased low to a degree, which can be estimated. This be discussed in more detail below. The alternative of operating at supersaturations low enough to ensure no droplet breakthrough would most likely also bias the immersion freezing ice-activated fraction low. This is because an unknown fraction of the sample aerosol potentially would not be exposed to supersaturations high enough to ensure activation into cloud droplets inside SPIN.

It is not straightforward to apply condensational growth calculations to an aerosol population inside a CFDC. Garimella et al. (2017) showed significant discrepancies between modeled and measured size distributions, which they ascribed to only a fraction of the sample being focused in the lamina.

We do not consider it necessary to carry out theoretical condensational growth calculations in this context, when the experimental reproducibility of single experiments is high. However, we consider it most likely, that even the largest surviving droplets to some extent shrunk in the evaporation section before detection.

We have not applied any changes to the manuscript in this context, since we find that the presence of the droplet mode already is significantly described e.g. in Fig. 2.

- L259: Why are you using 6 microns as size threshold for ice crystals, if you argue above that the cloud droplets do not grow larger than 4.5 microns? Do you not introduce another bias then to your reported activated fractions? Please clarify in the text.

A small tail of the droplet size distribution detected with the OPC would typically approach 6 microns in size at a sample temperature near -32°C with our operation conditions. Therefore, an ice threshold of 6 microns was applied throughout this study.

The text (L255-257) was changed from:

"Based on the homogeneous freezing experiments and all the other experiments we carried out under these conditions, the liquid droplets do not grow larger than approximately 4.5 μm in the SPIN when the sheath and sample flows of 9 LPM and 1 LPM, respectively, are used and the particle residence time in the IN chamber is 10 seconds."

To:

"Based on the homogeneous freezing experiments and all the other experiments we carried out with the same operation conditions, the very largest droplets detected with the OPC approached a diameter of 6 μm."

- L265: "since the ice crystal mode…" You might want to refer back to your Fig. 1 here.

Reference to figure added.

- L269: Please specify "in a later section"

The text has been changed from:

"The potential biases in the ice-active fraction associated with this approach are discussed in more detail in a later section."

to:

"The potential biases in the ice-active fraction associated with this approach are discussed in more detail in Sect. 3.4."

- L275: Delete "in real life"

Done.

- L276: Please clarify whether your ice counts are corrected by this factor. Which?

The presented ice-activated fractions are NOT exposed to any corrections. Since the correction factor is uncertain, which is inherent for CFDC experiments in general (Garimella et al., 2017).

- L280-283: "Ice onset is defined.." This statement should be moved to the description of the methods. Moreover, I find your terminology of "negative" and "positive" results quite misleading. For your T-scan experiments (fixed RH): In case you observe ice formation at T > -38°C (or your experimentally defined homogeneous freezing threshold), refer to it as heterogeneous ice formation. Similarly, in your RH-scan experiments that take place at T = -32 °C, you are above the homogeneous nucleation temperature, so any ice formation you observe should be heterogeneous.

The statement is moved to Methodology section and mentioned after the definition of the activated fraction.

The latter statement has been changed from:

"A result is considered positive if the ice onset has taken place under conditions when the sample temperature or *RHi* have been significantly higher or lower, respectively, than conditions required for homogeneous freezing. The result is considered negative if ice crystals have not been observed or if there is no distinguishable difference to homogeneous freezing conditions."
to:

"A result is considered positive if heterogeneous ice nucleation is observed under freezing conditions that are distinguishable from homogeneous freezing, and negative if ice crystals have not been observed or if there is no distinguishable difference to homogeneous freezing conditions."

- L286: Please define "ultrafine", also for usage further down, e.g. L329.

The text has been changed from:

"All samples were dominated by ultrafine particles by number,"

to:

"All samples were dominated by ultrafine (< 100 nm) particles by number,"

- L289-L291: You need to elaborate on various statements made here:

o The number concentration introduced into SPIN seems very high. Considering your aerosol to sheath flow ratio, you still have around 250 #/cc entering the OPC downstream of SPIN. What is the upper number concentration that can be sampled without co-incidence error?

Throughout the campaign we observed that all cookstove emissions were heavily dominated by ultrafine (< 100 nm) particles which go through the OPC undetected when operating SPIN without droplet breakthrough. Even with droplet breakthrough, only a fraction of the sample particles will be detected by the OPC – the ones focused in/near the lamina while also 'surviving' the evaporation section. At these particle concentrations, previous OPC characterisation showed no risk of co-incidence errors. Please see our response to also the next comment.

o It remains unclear how and why the detection limit depends on the particle number concentration introduced into the chamber, as you write in the caption of your Tab. 1. Could it be that at higher number concentrations introduced into SPIN you have relatively more particles > 250 nm (> 5%, as stated on L287) that become detected in your OPC? Please see also my comment to L210.

The detection limit is basically defined by the signal to noise ratio, in this context, the ice crystal counts relative to the background noise. Increasing the sample concentration thus may increase the signal and thus the sensitivity. Our operation with sample number concentrations at the order of $10^3$ cm$^{-3}$ should ensure negligible vapour depletion inside the CFDC (Levin et al., 2016)

o Your detection limit of the order α = 10-6 seems very low. It would be helpful to add an example plot for one of these experiments, showing α as a function of lamina RH in SPIN to the appendix. That way, the reader could more easily see at what RH you just sample noise and/or aerosol particles and at what point you actually start to see an ice signal. If α = 10-6 is your true detection limit, I suggest to more clearly write this in your statement on L211.

We have added a more detailed description of the detection limit to the original statement on L211, which is presented below:

"Typically, the background signal was below 1 particle per liter in the beginning of the experiment and the IN chamber was re-iced when it exceeded 10-15 particles per liter. This protocol, together with averaging the instrument data over 10-second periods as was done in data analysis, allowed detection of the ice-activated fraction from the order of $10^{-6}$ upwards in experiments where the sample concentration exceeded 2000 $cc^{-1}$. Data averaging enabled investigation of large droplet/ice crystal populations against their corresponding background signals, thus decreasing the lowest detection limit."

- L291: "Ice crystals…". This statement is not true looking at your "FD gasifier SW" experiment, if your threshold is α = 10-6.

Table 1 states that the detection limit in "FD gasifier SW" transient experiment was higher than $10^{-6}$ due to lower sample concentration.

- L291-294: Improve statement by saying: ""… four stoves do not nucleate ice in neither deposition nor immersion/condensation mode…"

Done.

- L297-318: This description should all be moved into your Sect. 2, where you describe your experimental methodology. Also, avoid unnecessary repetitions such as the mass concentration in the chamber (L304 vs. L122) to improve readability.

Description moved to Sect. 2.3.

- L323: Your statement on the soot sizes selected and your typical number concentrations differs from what is given on L129, please clarify. Furthermore, the number concentration of 50-200 #/cc significantly differs to the number concentration used for the transient experiments (see L290), please comment.

1. Size range and concentrations corrected to be consistent with L129.

2. The difference in number concentrations is due to size selection that was applied in chamber experiments, when only a fraction of the size selected particles makes it through the DMA. On the contrary, the transient experiments were done on polydisperse aerosol that was heavily dominated by ultrafine particles.

- L323: Please quantify the "minor contributions of multiply charged particles", so that the reader can judge on how your reported activated fractions are affected. See Wiedensohler (1988). I suggest adding sizes and fractions of double charged particles to your Table 2.

We have added the theoretical maximum estimate for doubly charged particle fraction to Table 2, following the calculations by Wiedensohler (1988). From there it can be clearly seen that the potential contribution of doubly charged particles account number concentrations that are much below the minimum reasonable sample concentration to the SPIN.

- L324: "...which indicates that soot…" So do you only consider particles > 250 nm as soot, please clarify. If so, what are all the other particles? Please see also my comment on L323, where you say that you investigated 200 nm particles.

The text has been changed from:

"which indicates that soot emissions were present in all studied cases."

to:

"which indicates that large soot particles were present in all studied cases."

- L329: Replace "when" by "after"

Done.

- L331: Replace "Chapter" by "Sect." Please read the guidelines of ACP here: https://www.atmospheric-chemistry-and-physics.net/for_authors/manuscript_preparation.html

- On the same note of abbreviations, please also use "Fig." instead of "Figure" throughout the text.

Done.

- L333: Change to "ice activation", to be consistent with e.g. L336. In any case, please check for consistent spelling throughout manuscript. Moreover, I feel this paragraph seems to refer to your Table 3, right? In this case, having a reference to this table at the beginning of this discussion might be helpful.

Yes, Table 3 supports the discussion. References added.

- L333: Please be more careful in specifying which of the 3S experiments you talk about. For instance, the 3S on 500 nm CAS (see Table 3), shows that the ice onset is detected at T = -37.8 °C, rather than at -38.1 °C, as you write on L333.

The number has been corrected to correspond to the ones in Table 3.

- L337: "It is not clear at which temperature…" This statement seems misleading. Is this not why you do your AS calibration results, to determine homogenous freezing conditions?

In order to model the theoretical freezing temperature based on classical nucleation theory, we would need to accurately know not only the time development of the droplet size distribution, respective temperatures and residence times for potentially different RH and temperature conditions. That is unknown for this and other CFDCs. And yes, that is why we determine the homogeneous freezing experimentally for the given operation conditions. The motivation for the comment, is that you can find examples in the literature of slightly higher homogeneous freezing temperatures closer to -38°C (e.g. Ignatius et al., 2016) potentially related to larger droplets.

The text has been changed from:

"The observed homogeneous freezing with $10^{-3}$ ice activation at an average lamina temperature of -38.9 °C may occur at slightly lower temperature than expected. It is not clear at which temperature homogeneous freezing in SPIN can be expected due to the sample aerosol being exposed to a span in temperature and RH, and uncertainties related to droplet sizes and residence times above the evaporation section as discussed by Ignatius et al. (2016)."

to:

"The observed homogeneous freezing with $10^{-3}$ ice activation at an average lamina temperature of -38.9 °C may occur at slightly lower temperature than expected from previous results reported in the literature. It is not clear at which temperature homogeneous freezing in SPIN can be expected from based alone on theoretical considerations due to the sample aerosol being exposed to a span

in temperature and RH, and uncertainties related to droplet sizes and residence times above the evaporation section as discussed by Ignatius et al. (2016)."

- L339-L341: Why do you talk about the biases of the activated fraction here, when above (L330) you say that this is discussed in Sect. 3.4. I suggest to move this to the discussion around L436. Please improve the structure.

Discussion on potential biases moved entirely to Sect 3.4.

- L342: Please elaborate how you estimate this correction activated fraction of 10-3 for the temperature interval indicated.

Application of a correction factor is mainly due to only a fraction of the sample particles being focused in the lamina and a potential presence of ice crystals below the threshold of 6 μm. Garimella et al. (2017) investigated the fraction of sample particles being focused in the lamina for SPIN. They reported 10% to 30% of the sample particles being focused in the lamina in most cases. That study indicates potential correction factors in the range from ~3 to ~10 to the ice-activated fraction. In our experiment, we estimate an upper correction factor to be ~8, which is based on the ice-activated fraction (for D>6 μm) reaching ~0.12 with the average lamina temperature being -41°C for the homogeneous freezing experiment. At that average lamina temperature, we would (in the ideal experimental setup) expect an ice-activated fraction approaching 1.

Application of correction factors in the range from 3 to 8 to the homogeneous freezing ice-activated fraction e.g. presented in Fig. 4 leads to a corrected ice-activated fraction of $10^{-3}$ in the temperature range from about -38.5°C to -38.0°C. Those homogeneous freezing temperatures are close to what we would expect from theoretical estimations as provided by Ignatius et al. (2016).

L342-343 have been changed from:

"We estimate that correction for those biases in the homogeneous freezing experiments will result in $10^{-3}$ ice activation between -38.5°C and -38.0°C in average lamina temperature."

to:

"Garimella et al. (2017) reported that the fraction of sample particles being focused in the lamina for comparable operation conditions of SPIN to be from 10% to 30% in most cases. In our homogeneous freezing experiment, the 'raw' ice-activated fraction as presented e.g. in Fig. 4 is about 0.12 for an average lamina temperature of -41°C. At that temperature, we would ideally expect an ice-activated fraction close to 1, which would be reached applying a correction factor of 8. For slightly higher lamina temperatures, the correction factor could potentially be lower, so we consider a correction factor in the range from 3 to 8 to be possible for this experiment for the lamina temperature in the range from -41 to -38.0°C. Application of such correction factors for the homogeneous freezing experiment will result in 10-3 ice activation between -38.5°C and -38.0°C in average lamina temperature."

- L349: Replace "as was done by" by "following"

Done.

- L349: "The uncertainty…" This sentence is a direct repetition of the previous statement, please delete.

Done.

- L353: Change "experiment emissions" to "experiments" and "high experiments" by "Good"

Done.

- L354: Change to "Sect. 3.4"

Done.

- L355: This statement needs to be clarified:

o I cannot infer any size dependence from your Fig. 4. In other words, all the experimental curves for the 3S experiments are within T-uncertainty of each other. I encourage you to also add α-uncertainty bars, maybe for every third data point for clarity/visibility.

The α-uncertainty bars that present the combined relative standard deviation (RSD) for the CPC and OPC will be added to the revised manuscript.

o Can you verify a difference in chemical composition between the runs using the SP-AMS, you indicate in your Fig. 1? If there is a chemical difference between the 3S CAS 300 nm and 500 nm experiment, it is meaningful to compare the ice nucleation activity of different sizes of these aerosol types?

The AMS analysis shows that the only difference in chemical compositions between the two mentioned experiments is higher BC to OA ratio in the 3S_CAS_300nm experiment, but we consider that it cannot explain the missing size dependency in IN activity. The differences of other studied physico-chemical properties were negligible in these two experiments. We would not expect to be able to identify physico-chemical differences between those experiments explaining the minor differences in ice-active observations – since our detailed analysis linking the results of more ice active samples to their properties presented in Figures 6 and 7, respectively, showed no clear pattern.

- L358: Add "than the ones…"

Done.

- L359: I cannot find the number "-37.7°C" in the RS experiments in your Tab. 3, please check.

Number corrected to -37.9°C.

- L361: "…and conditions for 10-3 ice-activated fraction were reached at -37.8 °C…" I think you do not need to repeat your ice nucleation onset activated fraction in every other statement. Just referring to "ice nucleation onset" would be fine (after you clearly defined it) and would improve the reading flow of your manuscript by a lot.

Corrected.

- L365: I would tune this down a little bit and say something along the lines of: "show that the soot particles emitted from burning the various fuels can act as heterogeneous INP at temperatures only slightly higher than those needed for homogeneous freezing of solution droplets."

Statement tuned down as suggested.

- L368-377: "This indicates that very minor changes in combustion conditions significantly influence the ice nucleating ability of freshly produced soot particles." I very much agree and I think this is a great finding, however, I think this paragraphs deserves some more clarifications:

o You say that you block one air supply, so your combustion should become less efficient, i.e. the organic carbon fraction of your soot particles should increase. Do you observe this in your SP-AMS data?

Unfortunately, there is not directly comparable SP-AMS data from the FD gasifier and SW fuel on normal operating conditions.

o On L371 you say that you varied the pot height (see also L92). However, it remains unclear, how this was changed, whether this was a systematic change and how this affects the combustion process:

The description of the modified experiments has been moved to Sect. 2.4, following guidelines given in general comments. The description of the "*pot height*" type experiments is as follows:

"The "*pot height*" experiments were done using different, less favorable heights of the cooking pot. The three such experiments were done on designated operation conditions of the FDGS and SW pellets, but the cooking pot was intentionally lifted to achieve production of larger particles: the real-time size distribution was monitored on the FPA throughout the experiment and the pot height was adjusted in a way that an increased production of large soot particles was observed. In typical cases, an offset of 8-10 cm above designated pot height affected production of large soot particles significantly."

o For instance, I find it interesting that you seem to see a very strong change in ice nucleation ability between "FDGS, mod. 400 nm #1" and "FDGS, mod. 400 nm #2" in your Fig. 6 (which you interpret as change in chemical composition due to different combustion conditions), but hardly no change between "FDGS, mod. 400 nm #1" and "FDGS, mod. 500n m". Can you explain this?

This could possibly be because AMS measures the (ensemble) average composition, while the IN constitute a very small subpopulation of the particles. It is not necessary that the size of that subpopulation co-varies with the average composition.

o On the same note there is a strong size dependence for the 450 nm and 500 nm FDGS samples with blocked secondary air supply, but no size dependence between the "FDGS, mod. 400 nm #2" and the "FDGS, mod. 500nm". Can you elaborate on this?

Unfortunately, the supportive data does not show which particle properties, or their combinations, could have caused this.

o Looking at your Fig. 7 it is also interesting to note that the FDGS samples with blocked are supply for the 450 and 500 nm case seem to be quite different in their properties, e.g. Org/BC. How reproducible are your soot test points? In the end, what looks like a size dependence in Fig. 6, might also be caused by differences in physico-chemical properties.

Here we do not agree, the B450 and B500 experiments both have high A370/A880 and high Cmid/Call. The difference in OA to BC is rather small (0.18 and 0.23).

- L382: Why is the effective density of 350 nm particles representative for your aggregates of different sizes? The effective density is a strong function of particle size, see e.g. Olfert et al. (2017), as you also note on L392.

Yes, the effective density is indeed a function of particle size, so we do not expect the effective densities for 350 nm mobility diameter particles to be identical to those for 400 or 500 nm particles. The effective density was measured for 350 nm particles in all the experiments relevant for the

results presented in Fig. 7, but unfortunately, we did not measure the effective density systematically for larger particle sizes.

Rissler et al. (2013) successfully applied 'simple' model fits to the effective density versus particle mobility diameter for different types of soot particles. If we assume a somewhat similar model of effective density vs mobility diameter to be applicable for our studied cases, then qualitative differences in the effective density for 350 nm particles should remain also for larger particle sizes. In other words, we do consider the differences in effective density for 350 nm particles to reflect differences in the effective density for the poly-disperse soot population to some extent.

This hypothesis is supported by a strong correlation (N=5, r=0.98, p=0.004) between the effective density for the 200 and the 350 nm particles within the soot mode for those 5 samples.

- L383: There is no Fig. 8.

Typing error corrected.

- L383-L391: This should be mentioned in your methods section along with a more detailed description and/or reference, how you calculate the chemical properties from the SP-AMS data, not in the results section.

Information that describes the method moved to methodology section.

- L391: "The values of…" Is this true for any soot aggregate size? Please specify.

No, this is not true for all soot aggregate sizes according to Rissler et al. (2013), to which we refer in that statement. They found that the values we mentioned were typical to soot aggregates larger than approximately 250 nm, i.e. the size range we sampled in our chamber experiments.

- L396: "OA" is not defined. You might also want to use "OA" instead of "Org" in your Fig. 7.

Yes, 'OA' is now defined when first introduced, and applied in figure label.

- L396: Should not K400#1 have a higher UV absorption, given the high Org carbon fraction?

At these low OA/BC ratios the AAE seems to be mainly driven by the components responsible for the refractory midcarbon signals. These have previously been associated with elevated AAE by Malmborg et al. (2019) and Török et al. (2018), which was stated later on L398-399.

- L412: Can you quantify this correlation?

The correlation was quantified on L404, stating that the $C_3O_2^+/C_3^+$ varied slightly between 0.005-0.007 from the least ice-active case to the most active one.

- L416: "Further studies…" I suggest to move this statement to your Sect. 4.

Moved to Sect. 4.

- L418: "The typical…" This is a repetition from L250 and can be deleted.

Repetition deleted.

- L425: Change "Chapter" to "Sect."

Done.

- L426: Replace "to" by "the"

Done.

- L428: This is a direct repetition of L346 and can be deleted.

Repetition deleted.

- L430-434: I think that it would be best practice to report the error corresponding to the largest uncertainty, i.e. the span in T across the lamina ("T profile") rather than the variation in mean lamina T. See e.g. Garimella et al. (2017)

- L441: What does "somewhat similar operation conditions" mean? You should clarify this.

The text has been changed from:

"somewhat similar operation conditions to what we apply in our study"

to:

"approximately similar operation conditions (T ≈ -33 °C at the lamina) to what we apply in our study"

- L445-461: This paragraph need to be significantly elaborated and improved in order to justify a proper discussion of biases in ice crystal detection.

o You list a bunch of reasons for underestimating you activated fraction (e.g. choice of 6 μm channel for ice detection), but you lack to quantify the contributions of this to the activated fractions reported here.

We have included α-uncertainty to activated fractions, so the reader can judge uncertainties in both lamina *T* and the particle detection.

o It remains unclear why the "underestimation depends on lamina temperature".

Yes, we agree that this statement may appear unclear. This underestimation depends on whether freezing has occurred in a size range significantly below our ice threshold as we observe strong indications of for an average lamina temperature near or below -40°C as explained in the following statement (L447-449). At such low sample temperatures, we observed more than one size mode to be dominated by ice, so we have also modified the formulation in the previous line (L446) slightly.

For the heterogeneous freezing results obtained well above the homogeneous freezing temperature, the vast majority of the ice crystal size mode appears above our size threshold of 6 microns, so this bias is much more pronounced for the lowest sample temperatures (near -40°C)

The text (L446-447) was changed from:

"only a fraction of the ice crystal mode is included in our reported ice active fractions. This underestimation depends on the lamina temperature."

to:

"only a (varying) fraction of the ice crystals were included in our reported ice active fractions. This underestimation turned significant when the sample temperature was low enough to induce freezing of 'small' droplets and/or possibly hydrated particles appearing at sizes well below the ice crystal threshold."

o How do you arrive at the upscaling factors of 3-5 and 5-10? Please add this to the text.

The lower correction factor estimate of 3-5 for $T_{lamina} \approx -33C$ is estimated by assuming at least a correction factor of 3 due to about one third of the sample aerosol being focused in the lamina (DeMott et al., 2015; Garimella et al., 2017). It may potentially be higher than 3, due to the potential presence of ice crystals with optical diameters <6 μm. In addition, potential losses of ice crystals between the chamber and the OPC have not been quantified. The higher estimated correction factors of 5-10 for $T_{lamina} \approx -41C$ are inferred in another manner. At this sample temperature, particles detected with the OPC in all size ranges (D>~0.7 μm) may be ice crystals, since we observe a clear transition in the average depolarization ratio in all size bins for $T_{lamina}$ going from -41 °C to -38 °C, which is indicative of a phase transition from ice crystals to liquid droplets/hydrated particles. We would also expect cloud droplets to freeze homogeneously at temperature close to -41 °C. However, the inferred ice-activated fractions as shown e.g. in Fig. 6, reach levels from about 0.08 to about 0.2, where we ideally would expect values close to 1. So we estimate the correction factors for $T_{lamina} \approx -41C$ to be at the order of 5-10, depending on the experiment. The statement has been improved from:

"DeMott et al. (2015) found that the ice active fraction should be upscaled by a correction factor of 3 for a CFDC and somewhat similar operation conditions to what we apply in our study. Garimella et al. (2017) reported upscaling factors for SPIN in the range from 1.5 to 10 depending on the operation conditions due to only a fraction of the sample aerosol being focused in the lamina."

to:

"DeMott et al. (2015) found that the ice active fraction for mineral dust particles should be upscaled on average by a correction factor of about 3 for a CFDC over a range of operation conditions including similar operation conditions (e.g. T ≈ -35 °C and $RH_w$ about 106-108% at the lamina) to what we apply in our study. However, for those specific operation conditions (lamina T=-35°C and just before droplet break through with an expected $RH_w$ about 106-108%) there is quite some scatter in the appropriate correction factor and a larger correction factor is often observed (DeMott et al., 2015). The offset can be ascribed to only a fraction of the sample particles being focused on the laminar flow. Garimella et al. (2017) reported upscaling factors for SPIN in the range from 1.5 to 10 depending on the operation conditions due to only a fraction of the sample aerosol being focused in the lamina. This is also evident in this study at laminar temperatures below -40 °C when all droplets are expected to freeze homogeneously: the inferred activated fractions (see Figs. 4-6) are between approximately 0.07 and 0.2, which suggests that a correction factor range suggested by Garimella et al. (2017) is required."

o Finally, please be more quantitative when saying "in a neighborhood of -40 °C"

The text has been changed from:

"The considerations discussed above lead us to estimate a likely upscaling factor at the order of ~3-5 of the ice active fraction for temperatures around -33 °C, while the upscaling factor is likely to approach 5-10 for temperatures in a neighborhood of -40 °C."

to:

"The considerations discussed above lead us to estimate a likely upscaling factor at the order of ~3-5 of the ice active fraction for lamina temperatures around -33 °C, while the upscaling factor is likely to approach 5-10 for temperatures significantly below homogenous freezing, around -40 °C."

- L468: "This study shows…" It remains unclear whether this statement refer to all different cook stoves tested or not.

The text has been changed from:

"This study shows that even small changes in combustion conditions can significantly affect the IN abilities of emission particles."

to:

"Based on observations from five special experiments on FDGS, this study shows that even small changes in combustion conditions can significantly affect the IN abilities of emission particles."

- L469: "500 million people…" This should be followed by a reference and I would move this statement to the introduction, as it does not constitute a conclusion from your study.

The text has been changed from:

"The regional daily usage by at least 500 million people suggests that the solid biomass burning"

to:

"Extensive regional usage of cookstoves suggests that the solid biomass burning"

- L471: "…showed good performance". This statement is misleading, given that your maximum activated fractions even for the AS experiments are 90% below the theoretically expected values, please be more specific.

The text has been changed from:

"The CFDC instrument in this study, the SPIN, showed relatively good performance and high reproducibility of experiments."

to:

"The CFDC instrument in this study, the SPIN, showed relatively good performance in temperature control, which enabled high reproducibility of experiments."

- L475: It would be good to quantify "poor IN activity" (not INP activity!) by giving a range of observed activated fractions.

The text has been changed from:

"We conclude on the experiments that the fresh, polydisperse emissions from cookstoves have a low INP potential at experimented temperatures (-28 °C, -32 °C). All emissions were heavily dominated by ultrafine particles that clearly showed poor INP activity. Moreover, their residence time in the atmosphere is relatively short due to deposition, coagulation and external mixing with other atmospheric particle species. Accumulation mode particles that were present in the transient experiments were not observed to activate heterogeneous ice nucleation at -32 °C in neither freezing mode, from which it can be concluded that the studied cookstove emissions have low IN activity at that temperature also in the immersion mode freezing."

to:

"We conclude that the studied polydisperse aerosol emissions from transient experiments with different cookstoves showed no indication of heterogeneous freezing above the detection limit (iceactivated fraction of about $10^{-5}$) for the investigated conditions (T=-28 °C and -32 °C and ~75-105% RHw). However, it should be kept in mind that those aerosol emissions typically were heavily dominated by ultrafine particles by number. Hence, the measurement sensitivity with respect to the accumulation mode particles alone was relatively low in those experiments. Therefore, we also carried out a number of immersion freezing experiments focused on size selected accumulation mode soot particles sampled from an aerosol storage chamber."

- L481: "Therefore…" Is this really true? Do all the cook stove emission be transported upwards? What about the atmospheric lifetime of these aerosols and what are the cloud properties that you say become changed?

We do not mean that all cookstove emissions end up in the upper atmosphere. However, a notable fraction of these PM emissions can be lifted to several kilometres in altitude in regions where convective draft is strong during daytime (viz. tropical and sub-tropical regions near the equator). The text has been changed from:

"Therefore, we conclude that usage of the cookstoves can emit potential INPs in the atmosphere, and thus affect cloud properties."

to:

"Therefore, we conclude that usage of the cookstoves can emit potential INPs in the atmosphere, and thus affect cloud properties such as the frozen fraction in MPCs and formation of cirrus clouds."

- L487-489: This comparison is very loose and insubstantial and requires a more adequate discussion and comparison of dust/soot emissions, (vertical) burdens, atmospheric lifetimes and many more factors. In the end, your results show that most of the soots can nucleate ice only very close to conditions required for homogeneous freezing. As such, the impact of these soot types on warm or MPCs is very likely absent and/or negligible. In fact, when one takes -38 °C as a "general threshold" for homogeneous freezing most of your soots in Figs. 4 and 5 freeze homogeneously

We have removed the comparison from the text.

- L490: Delete "of"

Done.

- L499: What do you mean by "slight differences"? Which of the "studied properties" were different? Please be more precise and quantitative here, otherwise it is hard for the reader to take out the main findings of your study.

The text has been changed from:

"Our analysis on physico-chemical properties of the emissions revealed slight differences in studied properties of soot particles that were present in the most ice-active results, yet these properties cannot define the IN efficiency alone."

to:

"Our supportive analysis on physico-chemical properties of the emissions revealed that hygroscopicity, OA-BC ratio, fraction of mid-carbon range fragments and fraction of refractory oxygen species cannot define the IN activity alone; the observed differences of these properties showed no clear trend towards increasing IN activity (see Fig. 7)."

- L761: Change "," to "." In front of "The two…"

Done.

- L767: The dependence of the detection limit on the sample concentration should be discussed in the main text. If this is the case, it would also be meaningful to add the concentrations for the individual experiments listed here.

- L779: Add "are equal…"

Done.

Bhandari, Janarjan, et al. (2019), 'Extensive Soot Compaction by Cloud Processing from Laboratory and Field Observations', *Scientific Reports*, 9 (1), 11824.

Bond, T. C., et al. (2013), 'Bounding the role of black carbon in the climate system: A scientific assessment', *Journal of Geophysical Research-Atmospheres*, 118 (11), 5380-552.

Brooks, S. D., Suter, K., and Olivarez, L. (2014), 'Effects of chemical aging on the ice nucleation activity of soot and polycyclic aromatic hydrocarbon aerosols', *J Phys Chem A*, 118 (43), 10036-47.

DeMott, P. J., et al. (2009), 'Ice nucleation behavior of biomass combustion particles at cirrus temperatures', *Journal of Geophysical Research-Atmospheres*, 114 (D16), D16205.

Ferry, D., et al. (2002), 'Water adsorption and dynamics on kerosene soot under atmospheric conditions', *Journal of Geophysical Research-Atmospheres*, 107 (D23).

Garimella, S., et al. (2017), 'Uncertainty in counting ice nucleating particles with continuous diffusion flow chambers', *Atmos. Chem. Phys. Discuss.*, 2017, 1-28.

Hoose, C. and Möhler, O. (2012), 'Heterogeneous ice nucleation on atmospheric aerosols: a review of results from laboratory experiments', *Atmospheric Chemistry and Physics*, 12 (20), 9817-54.

Kanji, Zamin A., et al. (2017), 'Overview of Ice Nucleating Particles', *Meteorological Monographs*, 58 (0), 1.1-1.33.

Koehler, Kirsten A., et al. (2009), 'Cloud condensation nuclei and ice nucleation activity of hydrophobic and hydrophilic soot particles', *Physical Chemistry Chemical Physics*, 11 (36), 7906-20.

Korolev, A., et al. (2017), 'Mixed-Phase Clouds: Progress and Challenges', *Meteorological Monographs*, 58, 5.1-5.50.

Lamb, Dennis and Verlinde, Johannes (2011), *Physics and Chemistry of Clouds* (Cambridge University Press).

Lohmann, Ulrike, Lüönd, Felix, and Mahrt, Fabian (2016), *An Introduction to Clouds: From the Microscale to Climate* (1st edition edn.; Cambridge: Cambridge University Press).

Mahrt, F., et al. (2018), 'Ice nucleation abilities of soot particles determined with the Horizontal Ice Nucleation Chamber', *Atmospheric Chemistry and Physics*, 18 (18), 13363-92.

Matus, A. V. and L'Ecuyer, T. S. (2017), 'The role of cloud phase in Earth's radiation budget', *Journal of Geophysical Research-Atmospheres*, 122 (5), 2559-78.

Mülmenstädt, Johannes, et al. (2015), 'Frequency of occurrence of rain from liquid-, mixed-, and ice-phase clouds derived from A-Train satellite retrievals', *Geophysical Research Letters*, 42 (15), 6502-09.

Olfert, Jason S., et al. (2017), 'Effective density and volatility of particles sampled from a helicopter gas turbine engine', *Aerosol Science and Technology*, 51 (6), 1-11.

Popovicheva, O., et al. (2008), 'Effect of soot on immersion freezing of water and possible atmospheric implications', *Atmospheric Research*, 90 (2), 326-37.

Pruppacher, H. R. and Klett, D. J. (1997), *Microphysics of Clouds and Precipitation* (2nd edition edn.; Dordrecht, The Netherlands: Kluwer Academic Publishers).

Wiedensohler, A. (1988), 'An approximation of the bipolar charge-distribution for particles in the sub-micron size range', *Journal of Aerosol Science*, 19 (3), 387-89.

---

## Author Comment (AC2) · 7 Feb 2020

**Author response to comments by Referee #2**

We thank Referee #2 for his/her work on the manuscript, and comprehensive comments and suggestions on revisions how to improve the contents. The detailed responses (written on red font) to comments as posted (green font) are listed below.

1. To better understand the implications of these measurements, it would be best convert the data shown in Figures 4 to 6 to active site density (ns) or active fraction kind of metric and compare against other INP data (soot, dust, etc.) from literature. This will help to put the data in the context of other INPs.

We agree that comparable metrics are essential when reporting scientific results. However, as we investigated the conversion options, we found that the active site density metric would very likely include big uncertainties due to assumptions (surface area) that are needed for such a conversion: biomass combustion soot particles have a tendency of being irregularly shaped agglomerates, unlike e.g. mineral dusts and/or biological particles such as bacteria. Therefore, when only their electrical mobility and effective density are known, the total surface area can range significantly from nearly spherical to a complex agglomerate yet maintaining the same electrical mobility and effective density. Hence, we consider results from this conversion speculative.

Instead, we shall present the data in activated fraction metrics and offer two-dimensional uncertainty analysis that contains uncertainties on lamina temperature (CFDC cooling system) and the statistical analysis of detector uncertainty comprising the OPC and the CPC.

2. Figures and Tables.

In Figure 1, do the ejector dilution (ED) also helps to cool the samples? (this is described on line 95).

Yes, because the compressed air we used was at room temperature. Regarding the high dilution ratio of 1:100, it can be expected that flue gas experienced also cooling during ejector dilution.

It is not clear how ice crystals can grow to size up to 11 um (Figure 2) as droplets only grew to 4.5 um only (page 256). Would you please explain this observation? If these droplets freeze, the size of the ice crystal should be equal to the droplet size, correct? From ~20.5 to 20.6 hrs (Figure 2), where ice crystals are observed, there are some particles of size 2±1 um observed. How is this possible? All the droplets should be frozen at this temperature. If these are not droplets, then why such small ice crystals are observed?

There are several reasons why the ice crystals appear significantly larger than the droplets detected with the OPC. In the following, we list two main reasons for the observed differences.

The detected particles had been exposed to an evaporation section where the droplets evaporated to some extent, while less is expected to happen to the ice crystals, so we do not know the exact size of the droplet population in the upper part of the chamber where freezing occurs.

The equilibrium vapour pressure is lower over an ice surface relative to a liquid water surface for temperatures below 0°C, so the growth of ice crystals is faster than the growth of liquid droplets in the chamber.

Please elaborate caption of figure 3. SPIN data is not shown here. How is the contribution from multiple charge particles is corrected for the data shown in figure 3?

The SMPS software includes automatic multiple charge correction, which was applied in the presented results in Fig. 3.

SPIN was operated at RHw =115% and without depol. detector. How was droplet breakthrough artefact addressed? It is not clear if the ice activation threshold is 1e-3, then how data is shown up to 1e-4(see figures 4 to 6). If the data (from 1e-3 to 1e-4) is not trustworthy because of the threshold limit, I would revise the figures to show data from 1e-3 to 1 only.

We addressed the droplet breakthrough artefact via choosing the ice crystal size threshold above the maximum droplet growth size (about 6 μm), which was determined from considering all experiments carried out in this manner during the campaign.

In chamber experiments, the threshold of $10^{-3}$ activated fraction does not represent the detection limit but the ice onset as we defined in the beginning of Sect. 3 on line 280: the purpose of this was to provide an ice activation threshold that is commonly used in the literature and is independent of measurement technique. The data between $10^{-4}$-$10^{-3}$ in ice-active fraction are well above the detection limit and thus trustworthy.

Please explain what X-axis labels in Figure 7 are. What is K500? How these labels are related to figures 4 to 6. I think there is a typo ('ja') on line 762. The ice detection limit (figure4 to 6) shows 1e-3, but in Table 1 detection limit is in the range of 1e-06. Please clarify this discrepancy and definition of the detection limit. From a readability perspective, it would be better to spell out the abbreviations (e.g., ND, FD etc.) that appeared in Tables 1 to 3 in the Table caption-text itself.

We have revised Fig.7 and tables in the following way:

- X-labels of Fig.7 have been changed to correspond with ones in Fig.6
- The typo on line 762 corrected
- Added definitions of ND and FD to caption of Table 3.

Please see the response to previous comment about the difference between detection limit and defined ice onset.

3. It is mentioned that the evaporation section is not efficient (line 262). Would you please explain this feature of the SPIN. Why it is not efficient, how it affects the data presented here, and how it is operated differently from other SPIN and CFDC style chambers. A paragraph from Line 273 to 276. Please elaborate on this argument. Does the correction factor was applied? If yes, how this factor was determined. There is some discussion in section 3.3; however, it is not very clear. The factor estimates described on line 458 to 461 are not proved using INP measurements. These are speculations. Please justify

Most of these issues have been addressed in our response to reviewer 1, and associated changes have been made to the manuscript. So here, we will only provide a few short comments.

The evaporation section of SPIN is shorter than for many other CFDCs, which most likely is due to a more advanced OPC in SPIN enabling detection of the depolarisation of the particles. Unfortunately,

the depolarisation signal detected with this particular OPC did not suffice for assessing the phase state of single particles.

All shown results were not exposed to any corrections of the ice-activated fractions, since we find the correction too uncertain to do so.

For the estimated correction factors, please see our response to reviewer #1 with respect to L445-461. Our estimates of corrections factors have been described in more detail in the revised version of the manuscript.

---

## Author Comment (AC3) · 7 Feb 2020

Dear Editor,

in addition to the revisions motivated by the review comments, we have also made a few other revisions to improve the quality of the manuscript. We have performed minor revisions to improve the language as well as a couple of more significant revisions. Those main revisions are related to (i) a more detailed description of the SP-AMS methodology, and (ii) the results presented in Fig.7 on supportive physico-chemical particle properties. The changes with respect to Fig. 7 are the following:

[Figure]

Ångström exponents. We have changed the Ångström exponents from being based on measurements at 2 wavelengths to include all 7 wavelengths in the range from 370 to 950 nm. That improvement did only result in very minor changes to the values reported – but these revised results are more robust.

Effective density of the "blocked sec. air 450nm" sample. During the quality control of all presented parameters, we discovered a mistake in how the effective density for the "blocked sec. air 450nm" sample was inferred. So, the effective density for that sample at 350 has consequently been changed from 0.215 to 0.252 gcm-3. The use of the correct value for that sample neither influences our discussion nor any of our conclusions.

---

## Author Response (AR1)

**Ice nucleating ability of particulate emissions from solid biomass-fired cookstoves: an experimental study**

Kimmo Korhonen[1], Thomas Bjerring Kristensen[2], John Falk[2], Robert Lindgren[3], Christina Andersen[4], Ricardo Luis Carvalho[3,a], Vilhelm  Malmborg[4], Axel Eriksson[4], Christoffer Boman[3], Joakim Pagels[4], Birgitta Svenningsson[2], Mika Komppula[5], Kari E.J. Lehtinen[1] and Annele Virtanen[1]

[1]University of Eastern Finland, Dept. Applied Physics. P.O. box 1627, FI-70211 Kuopio, Finland

[2] Lund University, Department of Physics, SE-22100, Lund, Sweden

[3]Umeå University, Thermochemical Energy Conversion Laboratory, SE-90187, Umeå, Sweden

[4] Lund University, Ergonomics and Aerosol Technology, Box 118, Lund SE-22100, Sweden

[5]Finnish Meteorological Institute, Atmospheric Research Centre of Eastern Finland, P.O. box 1627, FI-70211 Kuopio, Finland

[a] Now at: Centre for Environmental and Marine Studies, University of Aveiro, Department of Environment and Planning, PT-3810-193, Aveiro, Portugal

*Correspondence to*: Kimmo Korhonen (Kimmo.Korhonen@uef.fi)

**Abstract.** This research was part of the Salutary Umeå Study of Aerosols in Biomass Cookstove Emissions (SUSTAINE) laboratory experiment campaign. We studied ice-nucleating abilities of particulate emissions from solid fuel burning cookstoves, using a portable ice nuclei counter Spectrometer for Ice Nuclei (SPIN). These emissions were generated from two traditional cookstove types commonly used for household cooking in sub-Saharan Africa and two advanced gasifier stoves under research to promote sustainable development alternatives. The solid fuels studied included biomass from two different African tree species, Swedish softwood, and agricultural residue products relevant to the region. Measurements were performed with a modified version of the standard water boiling test on polydisperse samples from flue gas during burning and size-selected accumulation mode soot particles from a 15-$m^3$ aerosol-storage chamber. The studied soot particle sizes in nm were 250, 260, 300, 350, 400, 450 and 500. From this chamber, the particles were introduced to water-supersaturated freezing conditions (-32 °C to -43 °C) in the SPIN.

Accumulation mode soot particles generally produced an ice-activated fraction of $10^{-3}$ in temperatures 1-1.5 °C higher than that required for homogeneous freezing at fixed $RH_{water}$ = 115%. In five special experiments, the combustion performance of one cookstove was intentionally modified. Two of these exhibited a significant increase in the ice-nucleating ability of the particles, resulting in a 10⁻³ ice activation at up to 5.9 °C higher temperatures than homogeneous freezing and the observed increased ice nucleating ability. We investigated six different physico-chemical properties of the emission particles but found no clear correlation between them and increasing ice-nucleating ability. We conclude that the freshly emitted combustion aerosols form ice via immersion/condensation freezing at temperatures only moderately above homogeneous freezing conditions. ~~Ice nucleating abilities of particulate emissions from solid fuel burning cookstoves were studied using a portable ice nuclei counter SPIN (**SP**ectrometer for **I**ce **N**uclei) as part of the SUSTAINE (**S**alutary **U**meå **ST**udy of **A**erosols **IN** Biomass Cookstove **E**missions) laboratory experiment campaign. The emissions were generated from two traditional cookstove types commonly used for household cooking in sub-Saharan Africa, and two advanced gasifier stoves which are under research to promote sustainable development alternatives. The studied solid fuels studied included biomass from two different African tree species, Swedish softwood and agricultural residue products relevant to the region. Measurements were performed with a modified version of the standard water boiling test on 1) polydisperse samples from flue gas during burning and 2) size-selected accumulation mode (250-500 nm) soot particles from a 15 m³ aerosol storage chamber, from which the particles were introduced to water-supersaturated freezing conditions in the SPIN.~~

~~We observed that accumulation mode soot particles generally produced an ice-activated fraction of 10⁻³ in temperatures that were 1-1.5 °C higher than what was required for homogeneous freezing at fixed $RH_{water}$ = 115 %. Five special experiments where the combustion performance of one cookstove was intentionally modified were also performed, which led to a significant increase in the ice nucleating ability of the particles in two experiments, resulting in 10⁻³ ice activation at up to 5.9 °C higher temperatures than homogeneous freezing. Moreover, six different physico-chemical properties of the emission particles were investigated, but we did notfound no find a clear correlation between them and increasing ice nucleating ability. We conclude that in general, the studied freshly emitted combustion aerosols only facilitate immersion freezing at temperatures moderately above where homogeneous freezing occurs.~~

**1 Introduction**

Mixed-phase clouds (MPCs) play an essential role for climate and the hydrological cycle (Korolev et al., 2017; Mülmenstädt et al., 2015). Cloud droplets freeze homogeneously at temperatures near -38 °C (Pruppacher and Klett, 1997), but ice nucleating particles (INPs) may catalyze freezing of supercooled cloud droplets at higher temperatures. A wide range of MPC properties including radiative properties and lifetime are sensitive to the formation of solid ice (Matus and L'Ecuyer, 2017),  but there are significant gaps in our knowledge regarding INP concentrations within the MPCs and these important ice formation processes.

INPs active in immersion freezing at a temperature of  -30 °C are relatively rare in the lower troposphere with concentrations at the order of 0.01 cm⁻³ in many regions (DeMott et al., 2010). Ambient INPs include dust and biological particles and potentially soot particles ‑(Hoose and Möhler, 2012,; Kanji et al., 2017 and references therein). Soot particles from an acetylene burner, a kerosene burner and a soot generator have been reported to be active in immersion and condensation‑ freezing at temperatures up to ‑24°C‑24 °C (DeMott, 1990), ‑20°C‑20 °C (Diehl and Mitra, 1998) and ‑10°C‑10 °C (Gorbunov et al., 2001), respectively. However, a wide range of soot particles have been reported to be inefficient as INPs in immersion and condensation‑ mode (e.g. Hoose and Möhler, 2012), and the available parameterisations used to estimate the soot ice nucleating ability ‑parameterisations of the ice nucleating ability of soot particles span several orders of magnitude (Vergara-Temprado et al., 2018). The radiative forcing associated with the impact of fossil fuel soot particles on MPCs has been reported to range from about 0.1 to about 1 Wm$^{-2}$ depending on their ice nucleating ability, which is uncertain (Yun et al., 2013). The radiative forcing associated with the impact of soot particles on MPCs has been reported to be very uncertain and potentially up to about 1 Wm$^{-2}$ (Yun et al., 2013). The Intergovernmental Panel on Climate Change (IPCC) expressed in their latest assessment report (Boucher et al., 2013) a great need for additional research with respect to the role of soot particles in heterogeneous ice nucleation (Bond et al., 2013Boucher et al., 2013).

Soot particles are produced from incomplete combustion (combustion with insufficient oxygen supply) and ambient soot particle properties, such as morphology and chemical composition, are highly variable and influenced by the combustion conditions and atmospheric ageing (Ferry et al, 2002; Popovicheva et al., 2008; Koehler et al., 2009; Corbin et al., 2015; Mahrt et al., 2018; Bhandari et al., 2019). It is not entirely clear which soot particle properties influence the ice nucleating ability of the particles. Chemical groups on the soot particle surface with the ability to form hydrogen bonds with water molecules are likely to be of importance (Gorbunov et al., 2001) as well as the soot particle nanostructure (Häusler et al., 2018). Chemical groups on the soot particle surface with the ability to form hydrogen bonds with water molecules are likely to be of importance (Gorbunov et al., 2001). In addition, the soot particle nanostructure may also be of importance, with highly ordered graphene structures being more efficient in supporting ice nucleation relative to lowly ordered graphene structures (Häusler et al., 2018).

Biomass combustion for cooking is an important global source of energy, being also the major environmental health risk worldwide (Lim et al., 2012),. It has been estimated that approximately 2.8 billion people depend on solid fuel combustion for daily cooking worldwide, mostly in the developing part of the World (Lim et al., 2012; (Bonjour et al., 2013). Therefore, solid biomass combustion as it‑ is a significant source of particulate matter (PM) and soot particles on regional to global scales (Lim et al., 2012; Bonjour et al., 2013). However, studies of the associated ice nucleating ability are scarce. Ambient measurements indicate that biomass combustion is a source of INPs for MPC conditions (Twohy et al., 2010; McCluskey et al., 2014).‑. Detectable condensation/immersion freezing INP concentrations for a temperature of -30 °C have been reported for simulated wildfires in 9 of 21 and 13 of 22 experiments, respectively (Petters et al., 2009; Levin et al., 2016). Refractory black carbon has been observed to be associated with a significant fraction of the emitted INPs (Levin et al., 2016). Combustion with a modern log wood burner produced‑produced particles that acted as INPs at T = -35 °C, but not at T = -30

°C for condensation/immersion freezing (Chou et al., 2013). Huang et al. (2018) modeled the indirect climate impact of solid fueled cookstove aerosol emissions and  reported a potentially significant climate impact in increase of high clouds in their model runs where black carbon (BC) particles acted as INPs. Therefore, soot particle emissions can potentially affect MPC conditions and thus climate.

[revised manuscript text omitted]

Prior to experimentation, the fuels were pre-processed for the different types of cookstoves as follows: the fuel sticks used were chopped to pieces approximately 2 cm in diameter, 17 cm and 13 cm in length for the 3S and the RS, respectively, and dried at room temperature. The pellet fuels were pelletized at the TEC-lab into dimensions of 8 mm x 15 mm (diameter x length, respectively). Fuel usage was thus standardized for the repeated WBT experiments according to cookstove type. The 3S and the RS were loaded with 100 g of stick fuel in the beginning of each WBT, and re-fueled slowly and continuously throughout the cooking simulation to maintain the flaming combustion. A typical number of re-fueling events was between  five and ten. The gasifier stoves were loaded with 1 kg of pellets before lighting the fire, since they are batch-fired appliances where fuel is not  added during operation.

**2**.3 The CFDC instrument

Ice nucleation experiments were conducted with the spectrometer for ice nuclei (SPIN, DMT Inc., Colorado, USA), described by Garimella et al. (2016).

The main difference relative to the older version used by Ignatius et al. (2016) is improved temperature control. The main difference relative to the older version used by Ignatius et al. (2016) is improved temperature control via e.g. by an increase in the total number of thermocouples from 8 to 32 in the main chamber (growth section).

Briefly, the SPIN is a  (CFDC) instrument  using parallel plate geometry similar to the Portable Ice Nuclei Chamber (PINC) introduced by Chou et al. (2011) and the Zürich Ice Nuclei Chamber (ZINC) described by Stetzer et al. (2008). The sample flow at 1 standard liter per minute (SLPM), is sandwiched between two sheath flows of 4.5 SLPM each, and the residence time in the region where ice nucleation can take place is approximately 10 seconds. The diffusional flux of water vapor across the chamber is created via setting the ice-covered plates to different sub-zero temperatures, and the relative humidity with respect to water and ice ($RH_w$ and $RH_i$, respectively)

can be adjusted through the temperature difference between the plates. The aerosol is exposed to an isothermal evaporation section, upstream the detection with an optical particle counter (OPC). The temperature of the evaporation section was set to follow the temperature of the aerosol sample in the growth section of the chamber. The temperature, relative humidities and the ideal path of  the sample flow, from here on referred as lamina flow, are modelled according to a 1D flow model by Rogers et al. (1988). The ice layer was created to the IN chamber through cooling the walls to -32 °C  and filling the chamber with de-ionized water, which resulted in a thin layer of ice on the walls of the chamber. After filling the IN chamber was purged of excess water and vacuumed down to 70 mbar for 3 minutes, which removed loose ice and reduced roughness of the ice layer and thus the background signal. Here, the background signal refers to unwanted OPC counts in the ice crystal size range, which may be due to break up of frost on the ice-covered plates or alternatively due to tiny leaks in the instrument chamber. Typically, the background signal was below 1 particle per liter in the beginning of the experiment and the IN chamber was re-iced when it exceeded 10-15 particles per liter. This protocol, together with averaging the instrument data over 10-second periods as was done in data analysis, allowed detection of the ice-activated fraction from the order of $10^{-6}$ upwards in experiments where the sample concentration exceeded 2000 cm $^{-3}$ . Data averaging enabled investigation of large droplet/ice crystal populations against their corresponding background signals, thus decreasing the lowest detection limit.

 The (CPC ) that was operated parallel to the SPIN  monitor ed the sample concentration and thus provid ed required information for calculation of the ice-activated fraction. The ice-activated fraction $\alpha$ is  defined as:

$$\alpha = \frac{N_{ice}}{N_{CPC}} \qquad (1)$$

where $N_{ice}$ is the background-corrected concentration of ice crystals detected  by the OPC, considering the dilution resulting from aerosol to sheath flow within SPIN, and $N_{CPC}$ is the concentration of sample particles detected by the CPC. Background correction means that the frequency of background counts is linearly interpolated between background checks and the corresponding temporal values are subtracted from the measured signal. The uncertainty calculations in activated fraction assume that the OPC and CPC particle counts are randomly distributed, with the standard deviation then given by the inverse of the square root of counts, respectively. The total random uncertainty of $\alpha$ is based on the propagation of errors of the variables involved in $\alpha$ calculations (Eq. 1), namely IN counts, IN background before and after each run, and the total particle count. The depicted error bars represent ±1 standard deviation of the inferred random uncertainties.

We define the ice onset  as an ice-activated fraction of $10^{-3}$.  Freezing conditions with $RH_w < 100$ % at the lamina flow are named as deposition mode and freezing above water saturation is referred to as immersion freezing (Vali et al., 2015), because liquid droplet formation  is expected prior to freezing. Condensation and immersion freezing modes are indistinguishable in this instrument. The experiments were carried out using automated sequences that are available in this version of the SPIN, to ensure comparability between experiments on different fuels and cookstoves.

**2.4 Ice nucleation experiments**

The transient experiments were performed during the modified WBT using the transient sampling line, denoted A in Fig. 1. Fresh, polydisperse emission aerosol was introduced to the SPIN after desiccation to $RH_w$ < 5 %. The concentrations were diluted to the order of $10^3$ cm$^{-3}$ to avoid vapor depletion inside the SPIN. $RH$-scans were used for the transient experiments at constant T = -32 °C, scanning the $RH_i$ from 100% up to 160%, which is close to the procedure used by Petters et al. (2009) and Levin et al., (2016). The measurement sequence used in the transient experiments was as follows: The lamina $T$ was fixed and $RH_i$ was increased via broadening the temperature difference between the IN chamber plates. Eleven experiments were sampled transiently scanning at constant $T$ of -32 °C and ramping the $RH_i$ from 100 % up to 160 %, which is close to the procedure used by Petters et al. (2009). This scan included studying both deposition and immersion modes because $RH_w$ ranged from approximately 73 % to 115 %. Two experiment scans where T was fixed at $T$ = -28 °C °C had similar $RH_i$ range and the only difference was in higher lamina temperature. The experimental approach for the transient experiments is a standard operation procedure often applied in previous studies (e.g. Petters et al., 2009; Welti et al., 2009).

We introduced a different experimental approach with a focus on immersion freezing for the experiments involving sampling of combustion aerosol from the chamber. This procedure is illustrated in Fig. 2, where a homogeneous freezing experiment on highly diluted ammonium sulfate (AS, dry mobility diameter of 350 nm)) droplets was performed. The dry seed particles with a mobility diameter of 350 nm were introduced to the SPIN in number concentration of approximately 150 cm$^{-3}$., and droplets formed in the growth chamber of the SPIN due to operation with super-saturated conditions on water. This procedure is illustrated in Fig. 2, where a homogeneous freezing experiment on highly diluted aammonium sulfate is used as an example.

(AS, dry mobility diameter of 350 nm) droplets is used as an example. All experiments were carried out on similar automated $T$-scan ramps containing the following steps: First, the internal background was checked via sampling filtered dry air for at least 5 minutes before the lamina $T$-scanramp began. As $T$ is descending, homogeneous freezing causes detection of ice crystals at about -37.9 °C°C. When $T$ reaches the lowest set point of -43 °C °C the background is checked again via filtering sample flow at the inlet for 3 minutes, before opening it again and scanning the $T$ back to -32 °C °C (ascending ramp). The phase change back to droplets is detected at -38.9 °C°C, when the ice-activated fraction decreases to below 10$^{-3}$. The difference in ice onset and offset temperatures is most likely caused by cold pockets that may occur in the IN chamber during cooling in descending ramps. When the temperature control uncertainty is defined as one standard deviation from the averaged lamina temperature, it was observed that the temperatures typically deviate 0.9-1.0 °C °C and 0.2-0.3 °C °C from the set value of the lamina temperature (see the shaded area around lamina temperature graph in Fig. 2) in descending and ascending ramps, respectively. Therefore, all data obtained from the descending ramps were omitted from the analysis and only ascending ramps were studied. The total sampling time of about 2 hours, which was used in chamber experiments, enabled up to three repetitions for each WBT emission sample. This sampling time allowed 2-3  ascending $T$ scans for each chamber experiment, and the reproducibility of the experiments will be discussed in the Sect. 3.4.

Separation between liquid droplets and ice crystals was carried out using a basic particle size-threshold method in the immersion mode experiments. Based on the homogeneous freezing experiments and all the other experiments we carried out with the same operation conditions, the very largest droplets detected with the OPC approached a diameter of 6 µm

. This can be seen in Fig. 2 between 20.38-20.49 hrs and 20.75-20.87 hrs when the inlet is open, and the lamina $T$ is above homogeneous freezing temperature. Therefore, particles larger than 6 µm are considered ice crystals in all these experiments and the choice of this size threshold will be discussed in Sect. 3.4.

~~droplet mode. Lognormal distributions were fitted to the ice crystal size mode for different lamina temperatures higher than -36°C for the most ice-active sample studied in order to estimate the fraction of of ice crystals with optical sizes below 6 µm. We find that the ice-active fraction defined from the 6 µm threshold would have to be multiplied by a factor of 1.4±0.2 in order to obtain a reasonable estimate of the ice-activated fraction considering the entire ice crystal size mode – when the ice crystal mode was observed to be well-defined and ascribed to heterogeneous immersion freezing. The uncertainty range is~~

~~inferred from a sensitivity study including a range of reasonable fits, and it is due to noise in the data. It is not clear to which extent this correction factor is independent from the correction factor related to the sample aerosol not being entirely focused in the lamina. In other words, are the relatively smaller ice crystals (<6 µm) mostly representing a part of the sample not focused in the ideal lamina but rather in the vicinity of it? In any case, the bias associated with a relatively high ice size threshold is likely to be significantly lower than the expected bias due to only a fraction of the sample being focused in the~~

[revised manuscript text omitted]

It has previously been shown that the CFDC ice concentrations often are biased low by a factor of 3 and potentially up to a factor of 10 depending on the operation conditions [DeMott et al., 2015; DeMott et al., 2017; Garimella et al., 2017]. This bias is due to non-ideal behaviour of instruments , when the sample and sheath flows cannot follow the theoretical streamlines ideally. Hence, we will also estimate this effect and its possible effect on the results of this study.

**2.5 Emission particle characterization**

It is still largely an open question, which soot particle parameters influence the ice nucleating ability. However, the extensive online characterization of the aerosol applied in this work allowed us to investigate such potential links. The CCN activity, reported as the apparent hygroscopicity parameter $\kappa_a$, was inferred as described by Petters and Kreidenweis (2007) and the effective density $\rho_e$ in a similar fashion as done by Rissler et al. (2013). The aethalometer was used for studying optical 410 properties of the emission particles, measuring the Absorption Angstrom Exponent (AAE) in the wavelength interval from 370 nm to 950 nm. Furthermore, three physico-chemical properties were inferred from the SP-AMS measurements: organic aerosol to black carbon (OA/BC) ratio, relative abundance of refractory oxygen species and indirect information about the nanostructure of soot particles (Malmborg et al., 2019). The aim of this characterization was to study if the aforementioned six physico-chemical properties can explain variability in IN abilities from repeated experiments on different cookstoves, 415 fuels and combustion conditions.

**3 Results and discussion**

We present the IN results from transient and chamber experiments, compared to homogeneous freezing experiments whenever the experimentation included sample temperatures near or below homogeneous freezing temperature from the droplet freezing test.  A result is considered positive if heterogeneous ice nucleation is observed under freezing conditions that are distinguishable from homogeneous freezing, and  425  
[revised manuscript text omitted]
 statistical uncertainty of the activated fraction (vertical error bars) was found to be more prominent with low $\alpha$ values close to 10⁻⁴; in general, this uncertainty was at the order of 10-20% when the activated fraction was above 10⁻³, which agree with values reported by Garimella et al. (2017). The uncertainty analysis is based on modelled lamina temperature, the shaded area represents one standard deviation of lamina *T* from averaged lamina conditions at the time of each observation, when the 13 topmost thermocouple pairs are considered individually. The uncertainty analysies confirms that there is a significant difference in 10⁻³ onset temperatures between the 3S experiments emissions  and the homogeneous freezing experiment. HighGood experiment reproducibility (see Table 3) that will be discussed in Sect.ion 3.4 indicates very good agreement between repeated ramps and thus strengthens the validity of these observations. Size dependency that could be distinguished from the instrument uncertainty between 250, 300 and 500 nm particles was not observed, which may be is likely  due to differences in physico-chemical propertiescomposition between the individual experiments.

[revised manuscript text omitted]

Our detailed aerosol characterization presented in Fig. 7 indicates that we did produce particles with slightly different properties for intendedly identical experiments, as also was indicated by the observed difference in the ice nucleating ability. The low hygroscopicity, the low effective density and the optical properties indicate that the studied soot particles were not covered or coated by significant amounts of soluble material despite the presence of some organic compounds. It is
noteworthy that the two more ice active soot samples (450 nm on blocked secondary air supply and the first repetition of the *"pot height"* experiment with 400 nm) are significantly different with regards to several properties. The only minor visible trend  with ice-activity is a slightly lower ratio of refractory oxygen species ($C_3O_2^+/C_3^+$) for the two more ice active soot samples relative to the less ice active samples. However,  that trend is not significant in a direct comparison to the freezing temperatures, and the differences in refractory oxygen species between the
samples are not significant considering the errors. In general, these supportive data indicate that either (i) the inferred properties are not determining the ice nucleating ability, or (ii) a potential for complex combinations of different soot particle properties being of relevance for the ice nucleating ability. Additional studies are needed to address these questions in more detail.

**3.4 Experiment reproducibility and bias in ice crystal detection**

In this section, we discuss the reproducibility of the observations to validate the results and to evaluate the measurement precision of the SPIN. The results from repeated ramps are summarized in Table 3 where we present the ice onset temperatures and their uncertainty at the lamina $T$. Unlike in Figs. 4-6, we present the uncertainty as one standard deviation from the average lamina $T$ in this context, to show the overall precision of the temperature control. Observation numbers #1, #2 and #3 represent the IN onset after chamber residence times of approximately 20, 40 and 60 minutes, respectively, after filling the chamber was completed. These comparisons show that the results from repeated ramps are generally in good agreement with each other, and the differences in onset temperature remain within the typical deviation of the SPIN (see Sect. 2.3). It can be expected that the most prominent chamber effect has been coagulation of the ultrafine particles to large soot particles, but this has not affected the ice nucleating ability.

It should also be mentioned that across the width of the lamina, there is an additional span in temperature typically at the order of $\pm 0.4$ °C, so a fraction of the aerosol will be exposed to further lower or higher temperatures, respectively, compared to just the variability in average lamina temperature as presented in Fig. 2. However, we find that the variability in average lamina temperature is a reasonable estimate of the error in sample temperature for the experiments presented in Figs. 4-6 considering the high reproducibility of the results presented in Table 3.

Furthermore, the ice-active fraction is in the current study biased low for two reasons: (i) only a fraction of the sample aerosol is focused in the lamina (Garimella et al., 2017), and (ii) only the larger size fraction of the ice crystals (optical diameter >6 µm) is included in the ice-active fraction (Fig. 2). Garimella et al. (2017) reported that the fraction of sample particles being focused in the lamina for comparable operation conditions of SPIN to be from 10% to 30% in most cases. In our homogeneous freezing experiment, the 'raw' ice-activated fraction as presented e.g. in Fig. 4 is about 0.12 for an average lamina temperature of -41°C. At that temperature, we would ideally expect an ice-activated fraction close to 1, which would be reached applying a correction factor of 8 in that case. For slightly higher lamina temperatures, the correction factor could potentially be lower, so we consider a correction factor in the range from 3 to 8 to be possible for this experiment for the lamina temperature in the range from -41 to -38.0 °C. Application of such correction factors for the homogeneous freezing experiment will result in 10-3 ice activation between -38.5°C and -38.0 °C in average lamina temperature.

The presented ice active fractions are biased low for two main reasons as mentioned above. As it can be seen from Figs. 4-6, the ice active fractions for a temperature of -41 °C is typically at the order of 0.1 to 0.2, while we would expect an ice active fraction close to 1 for homogeneous freezing at that temperature below -40 °C. These observations confirm that the presented ice active fractions are biased low. Previous studies show that only a fraction of the sample aerosol is focused in the lamina. DeMott et al. (2015) found that the ice active fraction for mineral dust particles should be upscaled on average by a correction factor of about 3 for a CFDC over a range of operation conditions including approximately similar operation conditions (e.g. T ≈ -35 °C and *RHw* about 106-108% at the lamina) to what we apply in our study. However, for those specific operation conditions (lamina T= -35 °C and just before droplet break through with an expected *RHw* about 106-108%) there is quite some scatter in the appropriate correction factor and a larger correction factor is often observed (DeMott et al., 2015). The offset can be ascribed to only a fraction of the sample particles being focused on the lamina. Garimella et al. (2017) reported upscaling factors for SPIN in the range from 1.5 to 10 depending on the operation conditions due to only a fraction of the sample aerosol being focused in the lamina. This is also evident in this study at laminar temperatures below -40 °C when all droplets are expected to freeze homogeneously: the maxima of the inferred activated fractions (see Figs. 4-6) are between approximately 0.07 and 0.2, which indicates that correction factors in the range from 5 and up to 13 may be of relevance for our approach and these specific conditions at low temperatures.

In the current study, we have droplets and ice crystals coexisting after the evaporation section for a significant range of conditions, and with our established ice crystal threshold (optical diameter >6 μm), only a (varying) fraction of the ice crystals were included in our reported ice active fractions. The ice crystal size modes had maxima close to an optical diameter of 8 μm in the T-scan experiments, and a fraction of the ice crystal size mode would be present below 6 μm potentially overlapping with the droplet mode. Lognormal distributions were fitted to the ice crystal size mode for different lamina temperatures higher than -36°C for the most ice-active sample studied in order to estimate the fraction of of ice crystals with optical sizes below 6 μm. We find that the ice-active fraction defined from the 6 μm threshold would have to be multiplied by a factor of 1.4+0.2 in order to obtain a reasonable estimate of the ice-activated fraction considering the entire ice crystal size mode – when the ice crystal mode was observed to be well-defined and ascribed to heterogeneous immersion freezing. The uncertainty range is inferred from a sensitivity study including a range of reasonable fits, and it is due to noise in the data. It is not clear to which extent this correction factor is independent from the correction factor related to the sample aerosol not being entirely focused in the lamina.  In any case, the bias associated with a relatively high ice size threshold is likely to be significantly lower than the expected bias due to only a fraction of the sample being focused in the lamina.

The underestimation based on the size threshold of 6 µm turned more significant when the sample temperature was low enough to induce freezing of droplets smaller than 3-4 µm and/or possibly hydrated particles appearing at sizes well below the ice crystal threshold.  Investigation of the depolarization ratio of particles with an optical diameter in the sub-micrometer range with the SPIN OPC confirms that ice dominates in this size range at the lowest temperatures < -40 °C . Hence, the ice active fraction inferred from our procedure is underestimated, which is likely to be significantly more pronounced in the lowest temperature as can be seen from Fig. 2. Furthermore, the ice active fraction may also be underestimated due to (i) losses of ice crystals immediately above the OPC, which to our knowledge has not been studied in detail so far for SPIN, and (ii) a relatively small fraction of particles activating into droplets inside SPIN – leading to a reduced fraction for detection in the immersion mode. The potential losses have to our knowledge not yet been characterized in detail – while the ice-active fractions observed for homogeneous freezing of dilute ammonium sulfate droplets and droplets formed on hydrophobic particles typically resulted in very similar ice-active fractions (e.g. Fig. 6).

The considerations discussed above lead us to estimate a likely upscaling factor at the order of ~3-5 of the ice active fraction for lamina temperatures around -33°C, while the upscaling factor is likely to approach 5-10 for temperatures significantly below homogenous freezing, around ~in a neighborhood of~ -40 ~°C~ °C. Further studies are needed to assess these biases in more detail – and their relative importance for different operation conditions of the CFDC instrument.

**4 Summary and conclusions**

The SUSTAINE experiment campaign provided an excellent opportunity to study the IN efficiencies of emissions from different biomass-fired cookstove designs under well-controlled laboratory conditions, which enabled comparability between individual experiments. Two cookstoves, the 3S and RS, represented designs that are commonly used in daily household cooking in sub-Saharan Africa. The two more sophisticated designs, the NDGS and the FDGS, are currently under research for sustainable development and their popularity can be expected to increase along with economic development in this and similar regions. Based on observations from five special experiments on FDGS, ~T~this study shows that even small changes in combustion conditions can significantly affect the IN abilities of emission particles. ~The~Extensive regional ~daily~ usage of cookstoves ~by at least 500 million people~ suggests that the solid biomass burning emissions from cookstoves are a significant source of atmospheric particulate matter in sub-Saharan Africa. The CFDC instrument in this study, the SPIN, showed relatively good performance in temperature control, which enabled ~and~ high reproducibility of experiments.

the transient experiments were not observed to activate heterogeneous ice nucleation at -32 °C in neither freezing mode, from which it can be concluded that the studied cookstove emissions have low IN activity at that temperature also in the immersion mode freezing. We conclude that the studied polydisperse aerosol emissions from transient experiments with different cookstoves showed no indication of heterogeneous freezing above the detection limit (ice-activated fraction of about $10^{-5}$) for the investigated conditions (T= -28 °C and -32 °C and ~75-105% $RH_w$). However, it should be kept in mind that those aerosol emissions typically were heavily dominated by ultrafine particles by number. Hence, the measurement sensitivity with respect to the accumulation mode particles alone was relatively low in those experiments. Therefore, we also carried out a number of immersion freezing experiments focused on size selected accumulation mode soot particles sampled from an aerosol storage chamber." The results from chamber experiments on accumulation mode soot particles show that they can induce heterogeneous ice nucleation in higher temperatures than what is required for homogeneous freezing, as happened in most of the experiments. Therefore, we conclude that usage of the cookstoves can emit potential INPs in the atmosphere, and thus affect cloud properties such as the frozen fraction in low temperature MPCs. The chamber experiments also support the outcome of transient experiments that included the same cookstove-fuel combinations, because the experimental procedures included sampling at corresponding $RH_w$ of 115% but in lower temperatures. With these observations combined, we conclude that the fresh cookstove emissions that were tested both transiently and from the chamber do not contain components that are active INPs at or above a temperature of -32 °C. All but two chamber experiments show an obvious difference to homogeneous freezing experiment, and therefore these emissions may well be of relevance for INP number concentrations in their effect on atmospheric radiation budgetregions where other sources of ambient INPs are absent. Their atmospheric importance may be of second order in comparison to e.g. mineral dusts as Vergara Temprado et al. (2018) conclude in their study, but daily usage by hundreds of millions of people can still make cookstove emissions a significant regional source of atmospheric INPs. Despite of the chamber experiments consistently showing IN activity above homogeneous freezing temperature, it still needs to be noted that the studied emissions were relatively fresh: atmospheric ageing processes can affect the IN properties (Brooks, et al., 2014) before the emissions reach the upper atmosphere, as is being supported by Häusler at al. (2018). The effect of atmospheric ageing can be studied via using e.g. oxidation reactors in future studies.

Deterioration of combustion efficiency was observed to increase the INP potential of the emission particles, which is likely due to elevated large soot particle production in incomplete combustion of the tested biomass fuels. This, when combined with results from transient experiments and ones with standard combustion conditions, indicates that even modest changes in combustion efficiency can drastically alter the ice-nucleating capabilities of the emissions. Our analysis on physico-chemical properties of the emissions revealed that the soot particle non-refractory organic coating, carbon nanostructure, and light absorption characteristics were different between the least and the most ice-active results, yet none of these properties could define the IN efficiency alone. We recommend further studies that aim to investigate the relationships between the soot potential as INPs and soot nanostructure, refractory surface-bound oxygen, and non-refractory organic composition and coating thickness. It is clearly a possibility that the IN efficiency of soot is determined by a complex combination of multiple particle properties. It is therefore possible that co-correlated microphysical changes in the soot structure (on orders of 10-1000 nm) blurred the effects from changes in the non-refractory organic composition, nanostructure or oxygen bound to the soot surfaces.  supportive  that hygroscopicity, OA-BC ratio, fraction of mid-carbon-range fragments and fraction of refractory oxygen species cannot define the IN activity alone; the observed differences of these properties showed no clear trend towards increasing IN activity (see Fig. 7). ~~slight differences in studied properties of soot particles that were present in the most ice-active results, yet these properties cannot define the IN efficiency alone. Another possibility is that the IN efficiency of soot is potentially determined by a complex combination of multiple particle properties. We conclude this section that the inferred physico-chemical properties cannot determine the ice-nucleating ability of soot particles in this study. We recommend further studies for finding the relation between soot nanostructure and its potential as INPs.~~ Further studies are needed in order to reach any firm conclusion in this matter.

It is worth emphasizing that all experiments of this study were carried out in well-controlled laboratory conditions and using standardized test procedures. Contrary to that, real-life use can differ significantly from these experiments in many factors, such as in fuel pre-processing, fuel properties, technical stove conditions and practical cooking procedures. These all affect the combustion and emission performance that can have a prominent effect on ice-nucleating abilities of emitted aerosol particles, which this study shows. Therefore, the relevance between these observations and real-life use should be explored before the contribution of biomass-burning emissions can be further evaluated in global perspective.

**Data availability**

The data set is available upon request from Kimmo Korhonen (kimmo.korhonen@uef.fi).

**Competing interests**

The authors declare that they have no conflict of interest.

**Author contribution**

TBK planned the ice-nucleation experiments on the SPIN instrument that was operated by KK during the campaign. AV, JF, KK, MK and TBK participated in data analysis of ice-nucleation experiments and/or interpretation of results. RL and RLC prepared the experimental set-up for water boiling tests and operated the combustion facility during the campaign. AE, CA, JF, JP, TBK and VB-M participated in collection of supportive data and/or interpretation of results. BS, CB and JP participated as the organizers and supervisors of the SUSTAINE experiment campaign. AV and KEJL achieved funding for the SPIN instrument. All authors participated in scientific discussions on this study and reviewed/edited the manuscript during its preparation process. KK prepared the manuscript with contributions from all co-authors.

**Acknowledgements**

This work was financially supported by the Swedish Research Council FORMAS through the project Sustainable Biomass Utilization in Sub-Saharan Africa for an Improved Environment and Health (Dnr. 942-2015-1385) and Atmospheric cloud droplet formation and ice formation of wood combustion aerosols (2015-992). T.B. Kristensen gratefully acknowledges funding from the Swedish Research Council (VR) grant no. 2017-05016. LU-EAT researchers acknowledge financial support from the Swedish Research Councils VR (projects 2018-04200 and 2013-05021) and FORMAS (project 2013-

01023). University of Eastern Finland and Finnish Meteorological Institute acknowledge Academy of Finland, Centre of Excellence (grant no. 272041) and North-Savo Council - European Regional Development Fund's (project no. A32350). Ricardo Carvalho acknowledges the Postdoctoral grant JCK-1516 funded by the Kempe Foundation.

[Figure]

**Figure 1: Schematic of the experimental set-up used to conduct the WBT modified version relevant to the IN experiments. Sample lines A and B were used for transient and chamber experiments, respectively. Blue squares with letter "V" indicate valves which were used for switching between the sampling lines. The red and blue arrows show directions of sample flow related to transient and chamber experiments, respectively. The purple arrows show the flow direction used in both types of experiments, depending on position of the relevant switch valve. The acronyms are defined as follows: APM = aerosol particle mass analyzer, CCNC =**
**cloud condensation nuclei counter, CFDC = continuous flow diffusion chamber ice nucleus counter, CPC = condensation particle counter, DMA = differential mobility analyzer, ED = ejector dilution, FPA = fast particle analyzer, SMPS = scanning mobility particle sizer, SP-AMS = soot particle aerosol mass spectrometer. The dimensions and sample line lengths are not in scale to each other.**

[Figure]

**Figure 2: The immersion mode experiment procedure example from homogeneous freezing experiment with  350 nm ammonium sulfate seeds. Upper panel: Symbols T$_w$, T$_c$, and T$_a$ represent average temperatures for warm wall, cold wall and sample aerosol (lamina), respectively. The shaded black area in the top graph represents one standard deviation from the averaged lamina temperature. Lower panel: Events marked "BGD" depict background signal checks. The color bar indicates the logarithmic intensity of particle counts and the dashed black line the ice threshold size of 6 µm. The gap in detection near 3 µm is a**
**specific artefact in the OPC in this SPIN unit.**

[Figure]

**Figure 3: Examples of particle mobility size distributions from chamber experiments on different cookstoves, during the time when an activated fraction of $10^{-3}$ was observed in the SPIN.**

[Figure]

Figure 4: Ice-activation spectra of emissions from the 3-stone fire at $RH_w$ = 115%, chamber experiments. Each shaded area of respective color represents the $T$-span across the lamina (maximum uncertainty) ± one standard deviation on lamina temperature during each observation. The error bars present combined relative standard deviations in particle detection by the CPC and OPC of the SPIN. The solid black line presents the $10^{-3}$ activation threshold. The ice-activation spectrum for homogeneous freezing is included for comparison.

[Figure]

[Figure]

**Figure 5: Ice-activation spectra of emissions from gasifier stoves at $RH_w$ = 115% at regular operation conditions. Each shaded area of respective color represents the $T$-span across the lamina (maximum uncertainty) during each observation. The error bars present combined relative standard deviations in particle detection by the CPC and OPC of the SPIN. The solid black line presents the $10^{-3}$ activation threshold. The ice-activation spectrum for homogeneous freezing is included for comparison.** <s>Each shaded area</s>

of respectful color represents ± one standard deviation on lamina temperature during each observation. The solid black line presents the $10^{-3}$ activation. The ice-activation spectrum for homogeneous freezing is included for comparison.

[Figure]

[Figure]

**Figure 6: Ice-activation spectra of emissions from combustion of SW pellets in forced-draft gasifier stove, with modified combustion conditions at $RH_w$ = 115%. Each shaded area of respective color represents the $T$-span across the lamina (maximum uncertainty) during each observation. The error bars present combined relative standard deviations in particle detection by the CPC and OPC of the SPIN. The solid black line presents the $10^{-3}$ activation threshold. The ice-activation spectrum for homogeneous freezing is included for comparison.**

[revised manuscript text omitted]